# Convolutional networks can model the functional modulation of the MEG responses associated with feed-forward processes during visual word recognition

Marijn van Vliet[1]*, Oona Rinkinen[1], Takao Shimizu[1], Anni-Mari Niskanen[1], Barry Devereux[2], Riitta Salmelin[1,3]

[1]Department of Neuroscience and Biomedical Engineering, Aalto University, Espoo, Finland; [2]School of Electronics, Electrical Engineering and Computer Science, Queen's University Belfast, Belfast, United Kingdom; [3]Aalto NeuroImaging, Aalto University, Espo, Finland

*For correspondence:
w.m.vanvliet@gmail.com

Competing interest: The authors declare that no competing interests exist.

## eLife Assessment

van Vliet and colleagues show a **useful** correlation between internal states of a convolutional neural network (CNN) trained on visual word stimuli with three specific components of evoked MEG potentials during reading in humans. The findings are **solid**, though quantitative evidence that model can produce any of the phenomena that the human visual system is known to have (e.g., feedback connections, sensitivity to word frequency), or that it has comparable performance to human behaviour (i.e., similar task accuracy with a comparable pattern of mistakes) would make the conclusions much stronger.

**Abstract** Traditional models of reading lack a realistic simulation of the early visual processing stages, taking input in the form of letter banks and predefined line segments, making them unsuitable for modeling early brain responses. We used variations of the VGG-11 convolutional neural network (CNN) to create models of visual word recognition that starts from the pixel-level and performs the macro-scale computations needed for the detection and segmentation of letter shapes to word-form identification of large vocabulary of 10k Finnish words, regardless of letter size, shape, or rotation. The models were evaluated based on an existing magnetoencephalography (MEG) study where participants viewed regular words, pseudowords, noise-embedded words, symbol strings, and consonant strings. The original images used in the study were presented to the models and the activity in the layers was compared to MEG evoked response amplitudes. Through a few alterations to make the network more biologically plausible, we found an CNN architecture that can correctly simulate the behavior of three prominent responses, namely the type I (early visual response), type II (the 'letter string' response), and the N400m. In conclusion, starting a model of reading with convolution-and-pooling steps enables the flexibility and realism crucial for a direct model-to-brain comparison.

## Introduction

What computational operations is the brain performing when it recognizes some splotches of ink on a piece of paper as a meaningful word? This question has been the focus of a large number of neuroimaging studies that examine brain activity during reading. Noninvasive measurement techniques such

as electroencephalography (EEG; *Grainger and Holcomb, 2009*), magnetoencephalography (MEG; *Salmelin, 2007*), and functional magnetic resonance imaging (fMRI; *Price, 2012*) have provided a wealth of information about when and where changes in activity might be expected during various tasks involving orthographic processing (*Carreiras et al., 2014*).

In the case of visual word recognition, a lot of these results come in the form of changes in the amplitude of specific components of the neural response evoked by stimuli that are designed to create interesting experimental contrasts (*Luck, 2014*). Such evoked components reflect macro-level computations—that is, the net result of thousands of individual biological neurons working together. Taken together, the results indicate the presence of a processing pipeline, starting with the extraction of low-level visual features (e.g., edges, line segments), which are subsequently refined into more complex features (e.g., letter shapes) and further into lexical features (e.g., bigrams, words) (*Dehaene et al., 2005*; *Grainger and Holcomb, 2009*). While neuroimaging studies provide us with information about what processing steps are performed where and when, the observed data alone yield little information as to what kind of computations are performed during these steps (*Poeppel, 2012*). To develop such an understanding, we need to make these computations explicit, model them, and test and refine the model against the data provided by imaging studies (*Barber and Kutas, 2007*; *Price, 2018*; *Doerig et al., 2023*).

In this study, we aimed to computationally reproduce the results of *Vartiainen et al., 2011*, which is a representative MEG study that employed multiple experimental contrasts designed to study key processing steps throughout the first 500 ms of the visual word recognition pipeline. The authors catalogued the effects of the experimental contrasts on the amplitudes of all major evoked MEG responses found in the data, and concluded that the significant effects could be attributed to three components that dominate the early MEG time course during visual word recognition (*Salmelin, 2007*; *Jobard et al., 2003*), namely:

Type I: This component peaks occipitally around 100 ms after stimulus onset, is modulated by the visual complexity of the stimulus and hence thought to reflect the processing of low-level visual features (*Tarkiainen et al., 1999*). *Vartiainen et al., 2011* used a contrast between stimuli with and without added visual noise to highlight this processing stage.

Type II: This component peaks occipitotemporally around 150 ms after stimulus onset, and is modulated by whether the stimulus contains letters of the participant's native alphabet (*Tarkiainen et al., 1999*; *Parviainen et al., 2006*). Hence, this component is colloquially referred to as the 'letter string' response and thought to reflect letter shape detection. *Vartiainen et al., 2011* used a contrast between letter strings and symbol strings (visually similar to letters but not part of the Finnish alphabet) to highlight this processing stage.

N400m: This component peaks at around 400 ms after stimulus onset, and is often studied in the context of priming (*Kutas and Federmeier, 2011*), but experimental effects can also be observed when using isolated stimuli. For example, this component is modulated by the lexicality of the stimulus, hence associated with more lexical processing (*Halgren et al., 2002*; *Helenius, 1998*; *Service et al., 2007*; *Salmelin, 2007*). *Vartiainen et al., 2011* used a contrast between consonant strings, valid Finnish words and pseudowords to highlight this processing stage.

These three components are also observed in EEG studies (referred to as the P1, N1/N170, and N400 components, respectively; *Brem et al., 2009*). Together, they are indicative of three feed-forward processing stages during visual word recognition: basic visual analysis, orthographic analysis, and lexical analysis. To explore possible cognitive computations during these stages, we sought to create a computational model that performs the macro-level computations needed to achieve visual word recognition in such a manner as to reproduce the behavior of the three evoked components when presented with the same stimuli as a human participant.

In past decades, several computational models have been created that successfully capture some aspects of visual word recognition. For the purposes of this study, we draw a distinction between the older 'traditional-style' models (*Norris, 2013*) and the large artificial neural networks trained through deep learning that have been gaining popularity as models of brain function (*Richards et al., 2019*; *LeCun et al., 2015*). We will show that a model will need to combine elements from both approaches if it is to be able to reproduce the stimulus-dependent modulation of all evoked components mentioned above.

Among the first models of visual word recognition, the interactive activation and competition (IAC) model of letter perception by *McClelland and Rumelhart, 1981*; *Rumelhart and McClelland, 1982*, showed how the interplay of feed-forward and feed-back connections results in a system capable of 'filling in the blanks' when faced with a partially obscured word. This model was later extended to model semantics as well, showing how the activation of some semantic features ('is alive', 'is a bird') leads to the subsequent activation of more semantic features ('can fly', 'lays eggs'), in what became known as the parallel distributed processing (PDP) framework (*McClelland and Rogers, 2003*). Note that while models such as these consist of many interconnected units, those units and their connections do not aim to mimic a biological connectome, but are an abstract representation of macro-level computations performed by one. *Coltheart et al., 2001* pointed out the benefits of explicitly defined connections in their dual-route cascaded (DRC) model of visual word recognition and reading out loud, as they grant the researcher exact control over the macro-level computations. However, as the scope of the models increases, 'hand-wiring' the connections in the model becomes increasingly difficult. Therefore, most current models employ back-propagation to learn the connection weights between units based on a large training dataset. Together, IAC, PDP, and DRC models have been shown to account for many behavioral findings, such as reaction times and recognition errors, in both healthy volunteers and patients (*McLeod et al., 2000*; *McClelland and Rogers, 2003*; *Perry et al., 2007*). Furthermore, *Laszlo and Plaut, 2012* demonstrated how a PDP-style model can produce an N400m-like signal by summing the activity of the computational units in specific layers of the model.

However, none of the aforementioned models can produce detailed quantitative data that can be directly compared with neuroimaging data, leaving researchers to focus on finding indirect evidence that the same macro-level computations occur in both the model and the brain, with varying levels of success (*Jobard et al., 2003*; *Protopapas et al., 2016*; *Barber and Kutas, 2007*). The simulated environments in these models are extremely simplified, partly due to computational limitations and partly due to the complex interaction of feed-forward and feed-back connectivity that causes problems with convergence when the model grows too large. Consequently, these models have primarily focused on feed-back lexico-semantic effects while oversimplifying the initial feed-forward processing of the visual input. The models make use of 'letter banks', where each letter of a written word is encoded in as a separate group of inputs. The letters themselves are encoded as either a collection of 16 predefined line segments (*McClelland and Rumelhart, 1981*; *Coltheart et al., 2001*) or a binary code indicating the letter (*Laszlo and Plaut, 2012*; *Laszlo and Armstrong, 2014*). This rather high level of visual representation sidesteps having to deal with issues such as visual noise, letters with different scales, rotations and fonts, segmentation of the individual letters, and so on. More importantly, it makes it impossible to create the visual noise and symbol string conditions used in the MEG study to modulate the type I and II components. In order to model the process of visual word recognition to the extent where one may reproduce neuroimaging studies such as *Vartiainen et al., 2011*, we need to start with a model of vision that is able to directly operate on the pixels of a stimulus. We sought to construct a model that is able to recognize words regardless of length, size, typeface, and rotation with very high accuracy, while producing activity that mimics the type I, type II, and N400m components which serve as snapshots of this process unfolding in the brain. For this model, we chose to focus on the early feed-forward processing occurring during visual word recognition, as the experimental setup in the MEG study was designed to demonstrate, rather than feed-back effects.

Models of vision seem to have converged on a sequence of convolution-and-pooling operations to simulate the macro-level behavior of the visual cortex (*Serre et al., 2007*; *Lindsay, 2021*). Such models have become much more powerful as advances in deep learning and its software ecosystem are rapidly changing our notion of what is computationally tractable to model (*Richards et al., 2019*; *LeCun et al., 2015*). Convolutional neural networks (CNNs) have emerged that perform scale- and rotation-invariant visual object recognition at very high levels of accuracy (*Krizhevsky et al., 2017*; *Simonyan and Zisserman, 2015*; *Dai et al., 2021*). Furthermore, they model some functions of the visual cortex well enough that a direct comparison between network state and neuroimaging data has become possible (*Schrimpf et al., 2020*; *Devereux et al., 2018*; *Yamins and DiCarlo, 2016*).

Visual word recognition can be seen as a specialized form of object recognition (*Grainger, 2018*). Developmental studies suggest that orthographic-specific neuroimaging findings such as the visual word form area (VWFA) and the type II evoked components emerge after part of our established vision system is reconfigured as we learn to read (*Parviainen et al., 2006*; *Cohen and Dehaene, 2004*;

*Dehaene-Lambertz et al., 2018*). Studies on the visual cortex of non-human primates have shown that visual systems even without extensive exposure to written language have neural representations that carry enough information to distinguish letter and word shapes (*Rajalingham et al., 2020*), even accounting for distortions (*Katti and Arun, 2022*), and this finding was reproduced in CNNs. Therefore, CNNs may very well be suitable tools for increasing the scale of traditional connectionist models of reading, thus allowing for a much more realistic modeling of the early orthographic processing steps than what has been possible so far. Indeed, earlier work has established that training a CNN to classify written words can coerce it to form a VWFA-like region on top of an existing visual system (*Hannagan et al., 2021*), and cause it to form high-level orthographic representations (*Testolin et al., 2017*; *Katti and Arun, 2022*) which can be used by a linear regression algorithm to predict human MEG activity in the visual cortex (*Caucheteux and King, 2022*).

Whereas traditional-style models are typically designed to replicate a specific experimental effect in a biologically feasible manner, deep learning models are designed to perform in a more naturalistic setting, regardless of biological plausibility. Consequently, traditional-style models have been evaluated by comparing the timing and amount of the simulated activity to the amplitude of an evoked component (*Laszlo and Plaut, 2012*; *Laszlo and Armstrong, 2014*; *Nour Eddine et al., 2024*) or human reaction times (*Agrawal et al., 2020*; *Norris and Kinoshita, 2012*) in a tightly controlled experimental setting, and deep learning models have been evaluated using data from participants who are operating in a more challenging setting, such as watching movies (*Huth et al., 2012*), reading sentences (*Caucheteux and King, 2022*), or listening to stories (*Huth et al., 2016*). For the latter type of comparisons, information-based metrics are used. For example, representational similarity analysis (RSA; *Kriegeskorte et al., 2008*) quantifies the extent to which the inner state of a model discriminates between the same types of stimuli as a neural response, and BrainScore (*Schrimpf et al., 2020*) quantifies the extent to which a (usually linear) regressor or classifier can use the inner state of a model to predict brain activity. However, information-based metrics by themselves provide at best only a rough indication, and it is questionable whether better correlation scores truly indicates how closely a model mimics the computations performed by the brain (*Schaeffer et al., 2024*). Furthermore, if the score is not satisfactory, the metric does not provide a clear pathway to understand what the model is lacking and what could be done to improve it. Hence, it would be desirable to establish a more direct link between a model and the experimental results obtained in neuroscience studies (*Bowers et al., 2022*).

Therefore, in the current study, we evaluated the performance of a CNN in the spirit of the traditional models, namely by its ability to accurately reproduce the behavior of the type I, type II, and N400m responses in the tightly controlled experimental setting of *Vartiainen et al., 2011*. By doing so, we restrict ourselves to an investigation of how well the stimulus-dependent modulation of the three evoked components can be explained by a feed-forward CNN in an experimental setting designed to demonstrate feed-forward effects. As such, the goal is not to present a complete model of all aspects of reading, which should include feed-back effects, but rather to demonstrate the effectiveness of using a model that has a realistic form of input when the aim is to align the model with the evoked responses observed during visual word recognition.

We started with a basic CNN architecture, namely VGG-11 (*Simonyan and Zisserman, 2015*), that had been pretrained to classify images and evolved it into a model of visual word recognition by retraining it to classify written words. The performance of the model was evaluated by presenting it with the same set of stimuli that had been used by *Vartiainen et al., 2011* and comparing the amount of activity within its layers with the amplitude of the type I, type II, and N400m components measured from the human participants. Different versions of the model were created by introducing slight modifications to the base architecture and using different training regimens. These variations were first evaluated on their ability to replicate the experimental effects in that study, namely that the type I response is larger for noise-embedded words than all other stimuli, the type II response is larger for all letter strings than symbols, and that the N400m is larger for real and pseudowords than consonant strings. Once a variation was found that could reproduce these effects satisfactorily, it was further evaluated based on the correlation between the amount of activation of the units in the model and MEG response amplitude. The final model (1) has the ability of deep learning models to operate directly on the pixels of a stimulus, (2) can qualitatively replicate the experimental results of *Vartiainen et al., 2011* and some beyond, (3) has a large vocabulary of 10,000 words, and (4) yields

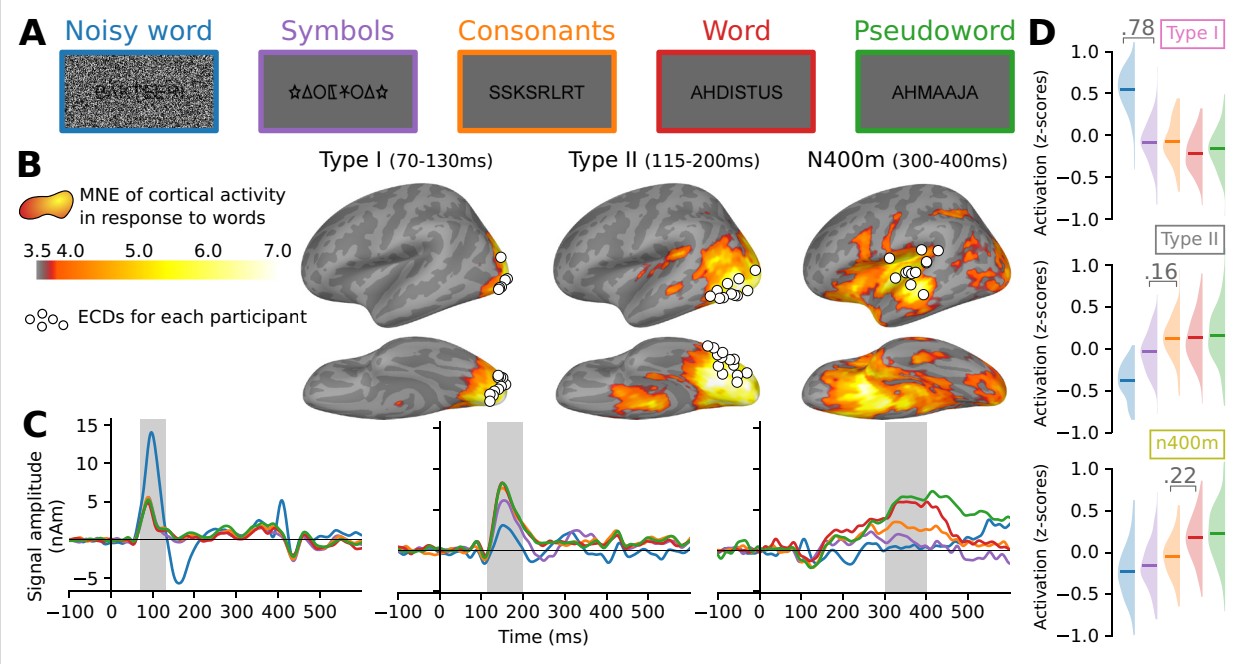

**Figure 1.** Summary of the magnetoencephalography (MEG) results obtained by *Vartiainen et al., 2011*. (**A**) Examples of stimuli used in the magnetoencephalography (MEG) experiment. Each stimulus contained seven to eight letters or symbols. (**B**) Source estimate of the evoked MEG activity, using MNE-dSPM. The grand-average activity to word stimuli, averaged for three time intervals, is shown in orange hues. For each time interval, white circles indicate the location of the most representative left-hemisphere equivalent current dipoile (ECD) for each participant, as determined by *Vartiainen et al., 2011*. (**C**) Grand-average time course of signal strength for each group of ECDs in response to the different stimulus types. The traces are color-coded to indicate the stimulus type as shown in (**A**). Shaded regions indicate time periods over which statistical analysis was performed. (**D**) For each group of ECDs shown in (**B**), and separately for each stimulus type (different colors, see **A**), the distribution (and mean) of the grand-average response amplitudes to the different stimulus types, obtained by integrating the ECD signal strength over the time intervals highlighted in (**C**). Whenever there is a significant difference (linear mixed effects [LME] model, p < 0.05, FDR corrected) between two adjacent distributions, the corresponding difference in means is shown.

The online version of this article includes the following figure supplement(s) for figure 1:

**Figure supplement 1.** Summary of the magnetoencephalography (MEG) results on the right hemisphere.

a good quantitative correlation between the amount of activation in various layers and the amplitude of the type I, type II, and N400m responses.

## Results

### The three MEG responses have unique response profiles indicative of different feed-forward processing steps

The computational models in this study were evaluated by their success in replicating the functional effects observed in the evoked MEG components as recorded by *Vartiainen et al., 2011* during the presentation of written words, pseudowords, consonant strings, symbol strings, and noise-embedded words (*Figure 1A*) to 15 participants. Performing distributed source analysis (*Dale et al., 2000*) yielded a comprehensive map of where activity can be found (*Figure 1B*), which is used in this study mostly to provide context for a deeper investigation into three peaks of activity that can be observed along the ventral stream. Through guided equivalent current dipole (ECD) modeling (*Salmelin, 2010*), the high-dimensional data was summarized as a sparse set of ECDs, each one capturing an isolated spatiotemporal component, allowing us to study selected components in more detail. The original study *Vartiainen et al., 2011* found three ECD groups along the ventral stream that correspond to the type I, type II, and N400m responses, based on their location (*Figure 1B*), timing (*Figure 1C*), and responses to the different stimulus types (*Figure 1D*). Similar ECD groups were identified in the right hemisphere (*Figure 1—figure supplement 1*), but the effects were clearer in the left. Therefore, in

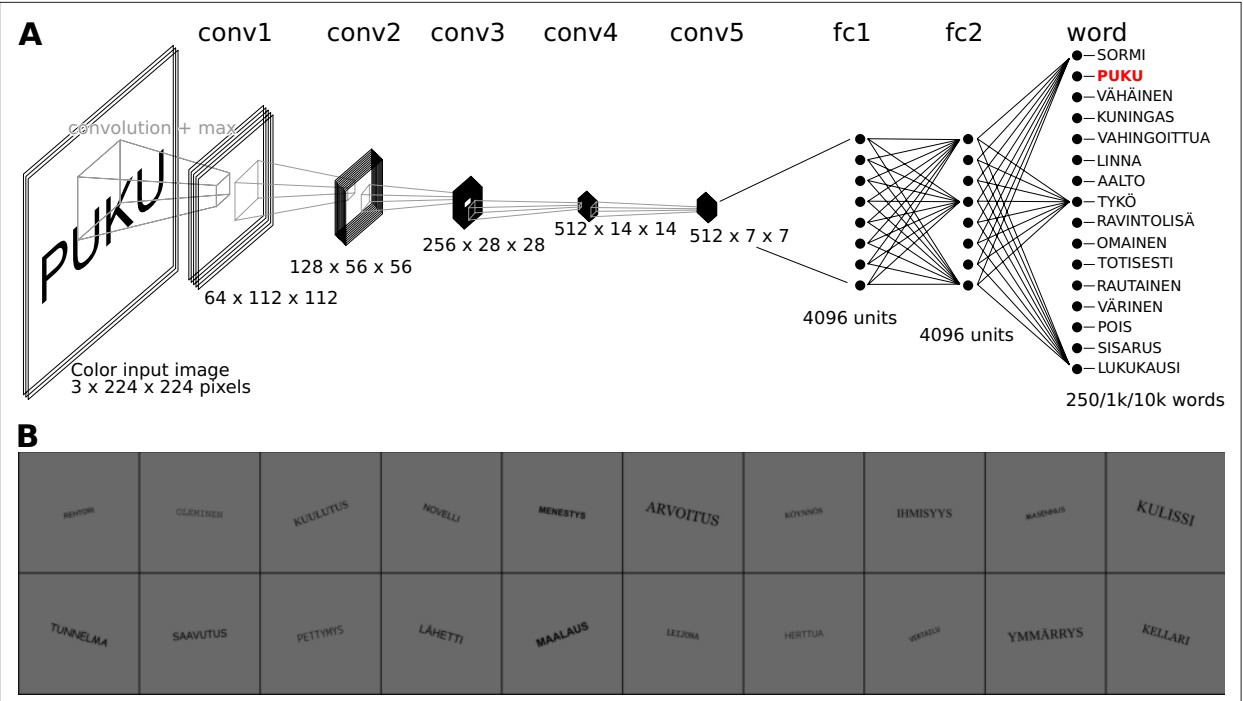

**Figure 2.** Overview of the proposed computation model of feed-forward processing during visual word recognition. (**A**) The VGG-11 model architecture, consisting of five convolution layers, two fully connected layers, and one output layer. (**B**) Examples of the images used to train the model.

this study, we re-used the three ECD groups on the left hemisphere to serve as ground-truth for our modeling efforts.

To compare the evoked MEG responses and model layer activity, we collected summary statistics of both, referred to as 'response profiles'. For the ECD time courses, the response profiles were formed by computing for each stimulus the average signal amplitude in the time intervals indicated in *Figure 1C*, z-transformed across the stimuli, and averaged across participants to obtain grand-average response profiles (*Figure 1D*). The response profile for the type I response demonstrates how this occipital component is driven by the visual complexity of the stimulus and is characterized by a large response to noise-embedded words relative to all other stimulus types. The type II response, located further along the fusiform gyrus, exhibits sensitivity to whether the stimulus contains letters that are part of the participant's native alphabet and, in contrast to the type I, has a reduced response to the noise-embedded words. The N400m response is located in the temporal cortex and is modulated by the lexical content of the stimulus, manifested in this study as a larger response to the word-like (i.e., words and pseudowords) versus the non-word-like stimuli. Taken together, these three response profiles are indicative of different phases during feed-forward processing, namely analysis of low-level visual features such as line segments and edges, followed by the analysis of letter shapes, and finally lexical analysis (*Salmelin, 2007*).

## CNNs can produce responses that are qualitatively and quantitatively similar to those of the three MEG evoked components

As a model of the computations underlying the brain activity observed during the MEG experiment, we started with a VGG-11 (*Szegedy et al., 2015*) network architecture, pretrained on ImageNet (*Russakovsky et al., 2015*). This architecture consists of five convolution layers (three of which perform convolution twice), followed by two densely connected layers, terminating in an output layer (*Figure 2A*). Variations of the model were trained to perform visual word recognition using training sets consisting of images of written Finnish words, rendered in varying fonts, sizes and rotations (*Figure 2B*). In each case, training lasted for 20 epochs, which was enough for each model to obtain a near perfect accuracy (>99%) on an independent validation set, as well as being able to correctly identify the original 118 word stimuli used in the MEG experiment. During training, the models were

never exposed to any of the other stimulus types used in the MEG experiment (noisy words, symbols, consonant strings, and pseudowords) nor to the exact word stimuli used in that experiment. After training was complete, response profiles for all layers were created by computing the $\ell2$ norm of the ReLU activations of the units in each layer in response to each stimulus, followed by a z-transformation (*Figure 3A*). Finally, the response profiles of the CNN models were compared to those of the three MEG evoked components by judging whether similar effects of the experimental contrasts could be observed. This allowed for an iterative design approach, where the network architecture and training diet of the model were manipulated to address shortcomings in the response profiles, until a model was reached with the response profiles of the contained layers matching those of the type I, type II, and N400m evoked MEG components.

Since both model layer activation and MEG evoked response amplitudes are described by a single number per stimulus in the response profiles, quantitative comparison between response profiles can be performed through straightforward correlation analysis (*Figure 3B*). While the models will consistently produce the exact same response to a given stimulus, evoked MEG response amplitudes can vary substantially between repetitions of the same stimulus, either when presenting it multiple times to the same participant or presenting it to multiple participants. The naturally occurring variability places a boundary on the maximal obtainable correlation score between model and brain responses, which we will refer to as the noise ceiling. The estimate for the single-participant noise ceiling, computed as the average correlation between the response amplitudes of a single participant versus those averaged across all remaining participants (*Kriegeskorte et al., 2008*), is very low for all three MEG components (type I: 0.236, type II: 0.112, N400m: 0.087), indicating a large amount of inter-participant variability. Therefore, we chose not to compare the models to single-participant data, but rather to the average across participants. The estimated noise ceilings for these average responses, as estimated through a procedure proposed by *Lage-Castellanos et al., 2019*, are much higher (type I: 0.765, type II: 0.618, N400m: 0.567), and hence the comparison will be more informative.

The design process started with an evaluation of the standard VGG-11 model, trained only on ImageNet. The layer response profiles of this model showed a distinctive pattern in response to the different types of stimuli used in *Vartiainen et al., 2011* (*Figure 3A*, top row), where the stimuli containing a large amount of noise evoke a very high level of activation in the first convolution layer, which carries over into all other layers. Thus, this model simulated the response profile of the type I component well, and correlation with the type I component was at the noise ceiling (*Figure 3B*, top), but it failed to simulate the response profiles of the type II and N400m components and correlated negatively with them. This is not surprising given that this model is illiterate.

Next, we trained the model on a training set containing a million examples of 250 possible written words (*Figure 3A*, second row). After the model had learned to recognize written words (regardless of size, font family, and some rotation), the convolution layers still showed a response profile similar to the type I. However, the later linear layers now had a less extreme response to the noisy stimuli and also showed a dissociation between symbols and letters, making their response profiles more similar to that of the type II and N400m components of the evoked response. Nonetheless, in this version of the model, the response to noisy stimuli is still too large to be a good fit with the response profiles of the type II and N400m.

In vision research, having unit activations be noisy has been shown to make a system more robust against image perturbations, both in the brain and in CNNs (*Dapello et al., 2020*; *Carandini et al., 1997*). In our case, the addition of a small amount of Gaussian noise to all unit activations in the model reduced markedly the response of the later layers to the noisy stimuli, resulting in less activity to the noisy stimuli than the other stimulus types (*Figure 3A*, third row), while the recognition accuracy of the model was not degraded. We found that in order for the model to display this behavior, both noisy unit activations and batch normalization were necessary (*Figure 3—figure supplement 1*). For this model, the response profiles of the first four convolution layers match that of the type I component, and the response profile of the last convolution layer is similar to that of the type II component in that both the noisy word stimuli and symbol stimuli evoked smaller responses than stimuli containing letters and all noiseless letter stimuli evoked comparable responses. However, this model does not contain any layers with response profiles matching that of the N400m. The N400m component of the evoked MEG response is modulated by the lexicality of the stimulus, as demonstrated through the contrast between consonant strings, proper words, and pseudowords. Whereas consonant strings

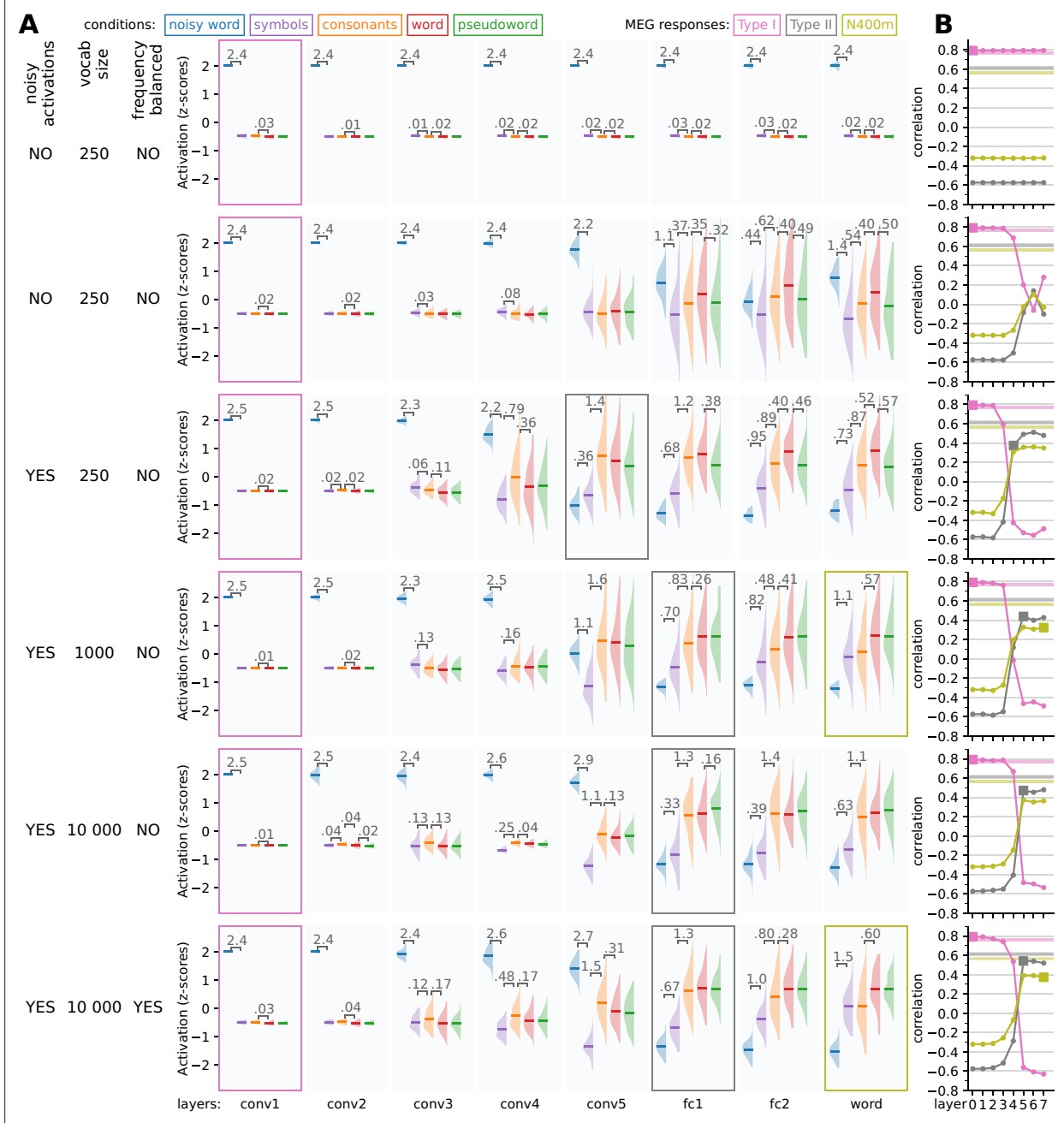

**Figure 3.** Building a model that can simulate the type I, type II, and N400m responses. Starting from a VGG-11 model, we made adjustments to the base architecture and training diet of the model to produce variations which simulated activity better matches the response profiles of the three magnetoencephalography (MEG) evoked components. (**A**) For each layer, the response profile, that is the z-scored magnitude of ReLU activations in response to the same stimuli as used in the MEG experiment, is shown. Whenever there is a significant difference (*t*-test, p < 0.05, FDR corrected) between two adjacent distributions, the corresponding difference in means is shown. Layers for which the response pattern was qualitatively similar to that of the type I, type II, or N400m component are outlined with a box of the appropriate color. (**B**) Correlation between the layers of each model (horizontal axis) and the three MEG evoked components (different curves). Layers for which the response profile (**A**) was judged to qualitatively correspond to one of the MEG components (*Figure 1D*) are indicated as filled squares. Noise ceilings for the MEG components are drawn as horizontal lines.

The online version of this article includes the following figure supplement(s) for figure 3:

**Figure supplement 1.** Impact of batch normalization and noisy activations.

**Figure supplement 2.** Impact of (pre-)training.

**Figure supplement 3.** Random initializations.

evoked a smaller N400m response than proper words, pseudowords produced a response that was equal to (or even larger than) that to proper words (*Figure 1D*). In this version of the model however, the response to pseudowords is less than that of proper words. This can be attributed to the relatively small vocabulary (250 words) of this model. The pseudowords used in the MEG experiment were selected to be orthographically similar to words in the extended vocabulary of a native Finnish speaker, but not necessarily similar to any of the 250 words in the vocabulary of the model. However, despite the failure to capture the N400m response profile, the fully connected layers of the model correlated much better with that component than the previous two models (*Figure 3B*, third row).

Increasing the vocabulary size to 1000 words was enough to solve the problem regarding pseudowords and yielded a model where the response profiles of the early convolution layers matched that of the type I, those of the two fully connected layers matched that of the type II, and that of the output layer matched that of the N400m (*Figure 3A*, fourth row). This did not cause a big increase in the correlation between the layers and the N400m (*Figure 3B*, fourth row), showing that a better qualitative fit of response profiles does not necessarily translate into a better correlation score.

Further increasing the vocabulary size to 10,000 words initially had a detrimental effect on the model's ability to simulate the N400m component, as the response to consonant strings was nearly as strong as that to words and pseudowords (*Figure 3A*, fifth row). In order to reproduce the response profile of the N400m during all experimental conditions of the MEG experiment, a model needs to hit a sweet spot regarding specificity of activation to in-vocabulary versus out-of-vocabulary letter strings. We found that this specificity can be controlled by manipulating the frequency in which words are presented to the model during training. By scaling the number of examples of a word in the training set in accordance with the frequency of occurrence of the word in a large text corpus (*Kanerva et al., 2014*), a model was obtained with a large vocabulary of 10,000 and the response profiles of the convolution layers matching that of the type I, response profiles of the fully connected layers matching that of the type II, and the response profiles of the output layer matching that of the N400m (*Figure 3A*, bottom row). These response profiles were stable across different random initializations when training the model (*Figure 3—figure supplement 3*). This manipulation of word frequency also led to a modest improvement in the correlation between the model and the type II and N400m responses. However, the main reason to include this variation was to achieve the desired response profiles while increasing vocabulary size. Compared to the model with a vocabulary of 1000 (*Figure 3B*, third row), the model with a vocabulary of 10,000 (*Figure 3B*, bottom row) achieved a substantially better correlation with the type II component, as well as a modest improvement in correlation with the N400m.

## The response profiles are dependent on the number of layers

Various variations in model architecture and training procedure were evaluated. We found that the number of layers had a large impact on the response patterns produced by the model (*Figure 4*). The original VGG-11 architecture defines five convolution layers and three fully connected layers (including the output layer). Removing a convolution layer (*Figure 4*, top row), or removing one of the fully connected layers (*Figure 4*, second row), resulted in a model that did exhibit an enlarged response to noisy stimuli in the early layers that mimics the type I response. However, such models failed to show a sufficiently diminished response to noisy stimuli in the later layers, hence failing to produce responses that mimic the type II or N400m, a failure which also showed as low correlation scores.

Adding an additional convolution layer (*Figure 4*, third row) resulted in a model where none of the layer response profiles mimics that of the type II response. The type II response is characterized by a reduced response to both noise and symbols, but an equally large response to consonant strings, real and pseudo words. However, in the model with an additional convolution layer, the consonant strings evoked a reduced response already in the first fully connected layer, which is a feature of the N400m rather than the type II. These kind of subtleties in the response pattern, which are important for the qualitative analysis, generally did not show quantitatively in the correlation scores, as the fully connected layers in this model correlate as well with the type II response as models that did show a response pattern that mimics the type II.

Adding an additional fully connected layer (*Figure 4*, fourth row) resulted in a model with similar response profiles and correlation with the MEG components as the original VGG-11 architecture (*Figure 4*, bottom row) The N400m-like response profile is now observed in the third fully connected layer rather than the output layer. However, the decrease in response to consonant strings versus

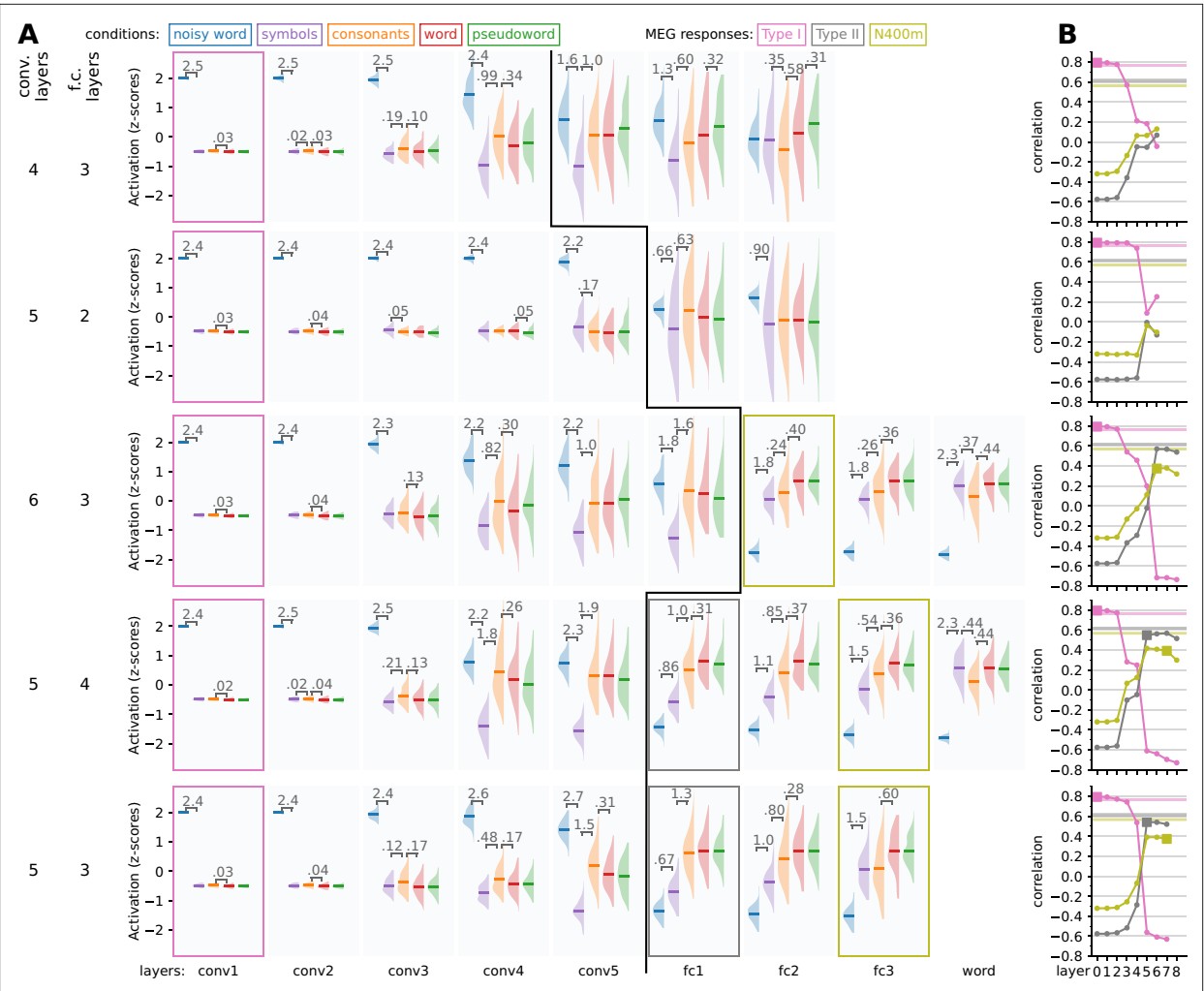

**Figure 4.** Exploring changes in model architecture. Model variations were constructed with a different number of convolution and fully connected layers to see what architecture produces activity that is the most like the three magnetoencephalography (MEG) components. (**A**) For each layer, the response profile, that is the z-scored magnitude of ReLU activations in response to the same stimuli as used in the MEG experiment, is shown. Whenever there is a significant difference (*t*-test, p < 0.05, FDR corrected) between two adjacent distributions, the corresponding difference in means is shown. Layers for which the response pattern was qualitatively similar to that of the type I, type II, or N400m component are outlined with a box of the appropriate color. A black line separates convolution layers from fully connected layers. (**B**) Correlation between the layers of each model (horizontal axis) and the three MEG evoked components (different curves). Layers for which the response profile (**A**) was judged to qualitatively correspond to one of the MEG components (*Figure 1D*) are indicated as filled squares. Noise ceilings for the MEG components are drawn as horizontal lines.

real and pseudo words, which is typical of the N400m, is less distinct than in the original VGG-11 architecture.

Some experimentation was done with the initialization of the model parameters. A model with either randomly initialized weights, or trained only on ImageNet, have response profiles in which each layer resembled that of the type I response (*Figure 3—figure supplement 2*, top rows). Training on word stimuli was required to simulate the type II and N400m components. While starting from a pretrained model on ImageNet reduced the number of epochs necessary for the model to reach >99% accuracy, pretraining was not found to be strictly necessary in order to arrive at a good model that can simulate all three components (*Figure 3—figure supplement 2*, bottom rows).

Based on our qualitative and quantitative analysis, the model variant that performed best overall was the one that had the original VGG-11 architecture and was pre-initialized from earlier training on ImageNet, as depicted in the bottom rows of *Figures 3 and 4*.

Various other aspects of the model architecture were evaluated which ultimately did not lead to any improvements of the model. The correlations between the models and the MEG components are

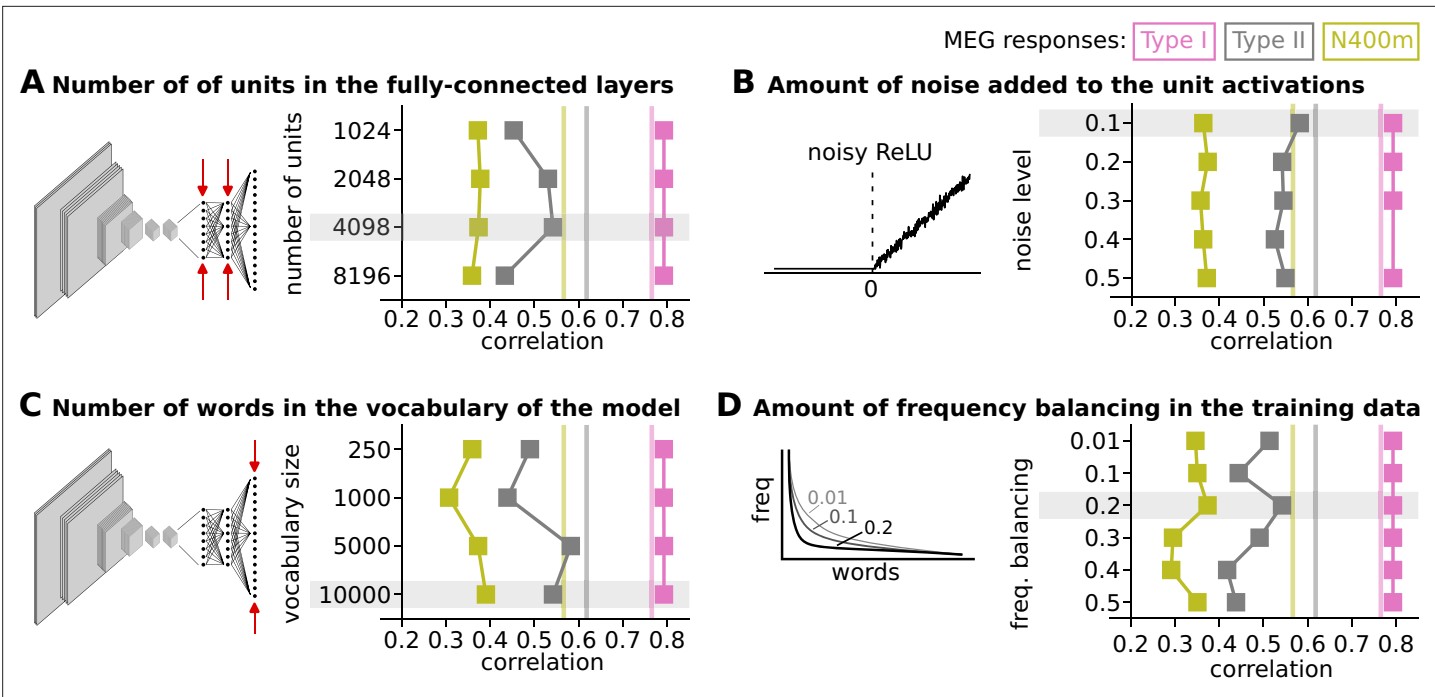

**Figure 5.** Impact of several hyperparameters on the correlation between model and brain. This figure shows the correlation between the response profiles of the type I, type II, and N400m evoked magnetoencephalography (MEG) components and the three layers in the models whose response profiles best match each of the MEG components. Estimated noise ceilings for each of the MEG components are shown as vertical lines. Each panel shows the impact of tweaking a hyperparameter and has an illustration to indicate the property of the model affected by the hyperparameter. The settings of the hyperparameters chosen for the final modal are highlighted in gray. (**A**) The number of units in the two fully-connected layers, the bottleneck, is being modulated. The impact of the number of units in the two fully-connected layers, the bottleneck, is being modulated. (**B**) The impact of the amount of noise ($\sigma_{noise}$) added to the activation of the units. (**C**) The impact of the number of words in the vocabulary of the model. (**D**) The impact of the amount of frequency balancing ($f^s$) in the training data.

The online version of this article includes the following figure supplement(s) for figure 5:

**Figure supplement 1.** Impact of fully connected layer width on model response profiles.

**Figure supplement 2.** Impact of amount of noise in the unit activations on model response profiles.

**Figure supplement 3.** Impact of vocabulary size on model response profiles.

**Figure supplement 4.** Impact of amount of frequency balancing on model response profiles.

presented in *Figure 5* and the corresponding response profiles can be found in the supplemantary figures of *Figure 5*. The vocabulary of the final model (10,000) exceeds the number of units in its fully connected layers, which means that a bottleneck is created in which a sub-lexical representation is formed. The number of units in the fully connected layers, that is the width of the bottleneck, has some effect on the correlation between model and brain (*Figure 5A*), and the amount of noise added to the unit activations less so (*Figure 5B*). We already saw that the size of the vocabulary, that is the number of word-forms in the training data and number of units in the output layer of the model, had a large effect on the response profiles (*Figure 3*). Having a large vocabulary is of course desirable from a functional point of view, but also modestly improves correlation between model and brain (*Figure 5C*). For large vocabularies, we found it beneficial to apply frequency balancing of the training data, meaning that the number of times a word-form appears in the training data is scaled according to its frequency in a large text corpus. However, this cannot be a one-to-one scaling, since the most frequent words occur so much more often than other words that the training data would consist of mostly the top-10 most common words, with less common words only occurring once or not at all. Therefore, we decided to scale not by the frequency $f$ directly, but by $f^s$, where $0 < s < 1$, opting for $s = 0.2$ for the final model (*Figure 5D*).

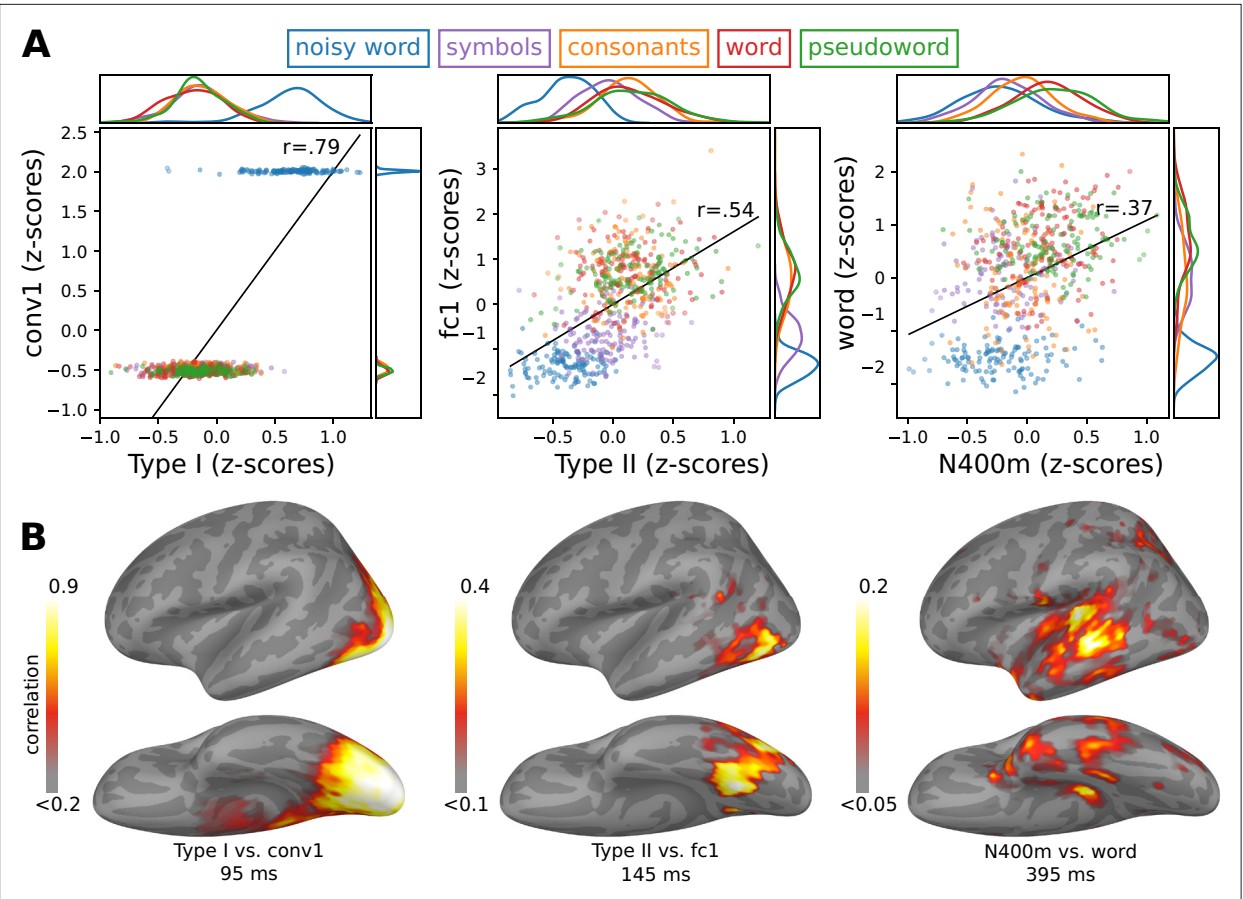

**Figure 6.** A closer look at the relationship between the final model and magnetoencephalography (MEG) responses. (**A**) A closer look at the relationship between the response profiles of the MEG responses and three layers of the model that qualitatively best capture those MEG responses. Kernel density distributions are shown at the borders. (**B**) Correlation between the MNE-dSPM source estimate and the model. Grand-average source estimates were obtained in response to each stimulus. The correlation map was obtained by correlating the activity at each source point with that for the chosen three layers of the model. The correlation map is shown at the time of peak correlation (within the time windows indicated in *Figure 1C*). Only positive correlations are shown.

The online version of this article includes the following figure supplement(s) for figure 6:

**Figure supplement 1.** Brainscore.

## Model–brain correlation is mostly driven by experimental conditions

Upon closer examination of the relationship between the response profiles of the final model and the MEG evoked components (*Figure 6A*), we see that the correlation is mostly driven by differences in response strength between the experimental conditions. Within each stimulus type, MEG evoked component amplitude and model layer activation are not significantly uncorrelated (p > 0.05 for all components), which is not surprising as the stimuli were designed to minimize within-condition variation. To verify that the three ECD groups capture the most important correlations between the brain activity and the model, we compared the model to cortex-wide MNE-dSPM source estimates (*Figure 6B*). The areas with maximum correlation appear around the locations of the ECDs and at a time that is close to the peak activation of the ECDs.

To test whether a multivariate analysis would yield additional insights, we trained a linear ridge regression model to predict brain activity given the activity within the layers of a model, what *Schrimpf et al., 2020* refer to as BrainScore. BrainScores where computed for both the illiterate model, trained only on ImageNet (*Figure 3*, top row), and the final model (*Figure 3*, bottom row). We found that for both models, every layer could be used to predict the brain activity along key locations along the ventral stream (*Figure 6—figure supplement 1*). Furthermore, there was very little difference in

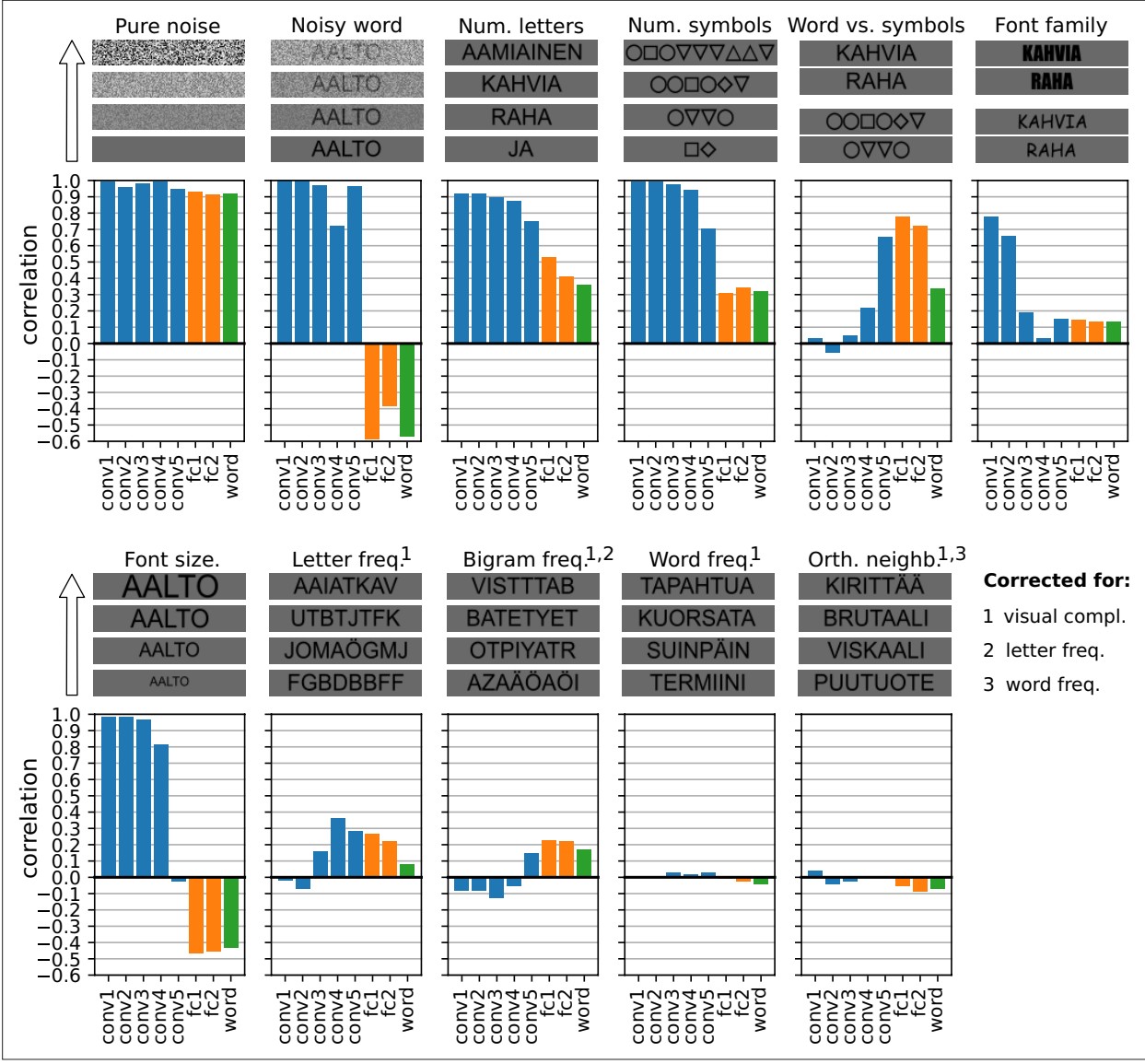

**Figure 7.** Post hoc exploration of various experimental contrasts. For each contrast, four sample stimuli are shown to demonstrate the effect of the manipulated stimulus property and below are the correlation between the manipulation and the amount of activity in each layer of the final model. For the experimental contrasts indicated with a number, one or more confounding factors were corrected for (partial correlation). Different colors indicate convolution layers (blue), fully connected layers (orange), and the output layer (green).

BrainScore between the illiterate and the final model. Hence, this information-based metric was in this case not helpful for evaluating model–brain fit.

## CNNs can reproduce experimental results beyond the original MEG study

The experimental contrasts employed in the MEG experiment were designed to distinguish the type I, type II, and N400m components from one another. However, many more experimental contrasts have been used to study these components in the past. We explored the effect of some of these on our final model (*Figure 7*) in order to ascertain the extent to which the model can simulate the behavior of the three components under conditions extending beyond the original MEG data that guided its design.

In *Tarkiainen et al., 1999*, the difference between the type I and II component was highlighted by contrasting stimuli that contained only noise with stimuli that contained both noise and letters.

Our model successfully replicates the finding that the amplitude of both the type I and II components increase with the amount of visual noise, except when there are letters present, in which case the amplitude of the type II component decreases with the amount of noise while that of the type I component is unaffected. This striking change in behavior occurs in the model when the layer type switches from convolution to fully connected linear. Furthermore, our model reproduces the finding by *Tarkiainen et al., 1999* that the amplitudes of both the type I and II component increases when a stimulus contains more letters or symbols, however given two strings of equal length, the amplitude of the type II component is larger for the string containing letters versus a string containing symbols, while the amplitude of the type I component is similar for both strings.

Basic visual features such as font family and font size have a large effect on the activity in the initial convolution layers of the model, and less so on the activity in the later layers. This roughly corresponds to findings in the masked repetition priming EEG literature, where they find that early components such as the N/P150 are affected by such properties, whereas later components are not (*Chauncey et al., 2008*; *Grainger and Holcomb, 2009*; *Hauk and Pulvermüller, 2004*). However, our results may not be directly comparable to that literature, because our model does not simulate repetition priming. For example, the fully connected linear layers in the model show a decrease in activity with increasing font size, which is an effect not found during repetition priming (*Chauncey et al., 2008*), but was found in studies that manipulated the font size of words presented in isolation (*Bayer et al., 2012*; *Schindler et al., 2018*).

Using invasive recordings, *Woolnough et al., 2021* found effects of letter and bigram frequency in the occipitotemporal cortex, where we find the type II component in MEG, but not in occipital cortex, where we find the type I component. This result has been shown in EEG studies as well (*Laszlo and Federmeier, 2014*; *Grainger, 2018*). Our model shows similar sensitivity to letter and bigram frequency in the later convolution and fully connected layers. However, we found that our model did not show any sensitivity to word frequency or orthographic neighborhood size, even though these are known to affect later components in EEG studies (*Holcomb et al., 2002*; *Dambacher et al., 2006*).

## Discussion

In this study, we demonstrated how training a CNN for visual word classification results in a model that can explain key feed-forward functional effects of the evoked MEG response during visual word recognition, provided some conditions to enhance biological realism are met. Necessary attributes of the CNN and its training diet were found to be that unit activations should be noisy, the vocabulary should be large enough, and for very large vocabularies, the frequency of occurrence of words in the training set should follow their frequency in the language environment. Following a neuroconnectionist approach (*Doerig et al., 2023*), we started with an illiterate CNN, trained to perform image classification, and iteratively made changes to its architecture and training diet, until it could replicate the effects of manipulating low-level visual properties such as noise levels, fonts and sizes, to mid-level orthographic properties such as symbols versus letters, and to high-level lexical properties such as consonants strings versus pseudowords versus real words. Some of these contrasts explore what we consider to be 'edge cases' of visual word processing that explore what happens when the brain tries to recognize a word and fails. Symbol strings, consonant strings, and pseudowords do not occur often in our natural environment and we do not explicitly train ourselves to distinguish them from real words when we learn to read. Following this logic, we included only real words in the training set for the model, so that responses to the non-word stimuli were simulations of a computational process failing to properly recognize a word.

### A computational process exhibiting type I, type II, and N400m-like effects

In this study, the computational process of visual word recognition was taken to be a transformation from a bitmap image containing a written word (with some variation in letter shapes, size, and rotation) to the identity of the word. Many network architectures can be found in the deep-learning literature that could by used to implement such a process. For this study, we chose the vgg architecture (*Figure 2*), where high accuracy is achieved by performing the same few operations many times, rather than many different operations or clever tricks. The operations performed by the network are:

convolution, pooling, batch normalization, and linear transforms. Our rationale was to start with these operations, which already have links to the neuroscience of vision (*Carandini et al., 1997*; *Serre et al., 2007*; *Yamins and DiCarlo, 2016*; *Lindsay, 2021*) and only when they prove insufficient to explain a desired experimental effect, introduce others. This turned out to be unnecessary for explaining the results of *Vartiainen et al., 2011*, but one can imagine the necessity of introducing recurrence (*Kubilius et al., 2019*) or attention mechanisms (*Vaswani et al., 2017*) in future extensions of the model into for example semantic processing.

The visual word recognition task itself did not prove particularly difficult for a CNN, as all variations of the network architecture and training diet that were tried resulted in a model that achieved near perfect accuracy. However, not all variations produced response profiles matching those of the evoked components (*Figure 3*). This validates a criticism sometimes heard concerning whether deep learning models are brain-like, namely that the ability of a model to perform a task that once could only be done by a brain is not enough evidence that a model computes like a brain (*Bowers et al., 2022*). Indeed, we found that task performance alone, that is the identification of the correct stimulus word, was a poor indicator for judging a model's suitability as computational hypothesis for the brain. More compelling evidence was obtained by comparing the model's internal state to measurements of brain activity. Two variations of the model, with respective vocabulary sizes of 1k and 10k words, implement the computational process in such a way that they produce amounts of activity within their layers that resemble the amplitudes of the type I, type II, and N400m components of the MEG evoked response. The model with the larger vocabulary was chosen to be the final iteration of the model and subjected to a more extensive range of experimental contrasts (*Figure 7*).

The computational process implemented by the final model starts with a series of convolution-and-pooling operations. Individual convolution units have a restricted receptive field, which is gradually widened through pooling operations. The units in the first layer see a patch of $3 \times 3$ pixels and those in the final fifth convolution layer combine information from patches of $18 \times 18$ pixels. This setup enforces a hierarchical representation forming in those layers, from edges to line segments and corners to (parts of) letter shapes (*Dehaene et al., 2005*, *Dehaene-Lambertz et al., 2018*). Hence, increasing the visual complexity, that is adding more edges to an image by for example introducing visual noise or adding more symbols, will increase the activity of the convolution units. This behavior corresponds closely to that of the type I component, which is in line with the growing body of evidence of convolution-and-pooling being performed in visual cortex (*Lindsay, 2021*; *Serre et al., 2007*).

The kind of information selected for in the later layers of the model changes through the introduction of non-relevant input signals during training, here done through noisy unit activations. Without noisy activations, the task of the computational units is limited to finding features to distinguish one letter from another. With noisy activations, their task also entails isolating letter-shape information from non-relevant information. The property of not merely having a preference for, but actively seeking to isolate certain types of information is key for producing the response profiles of the type II and N400m components. When receptive field sizes were $8 \times 8$ pixels and larger (fourth and fifth layers), the convolution units showed a preference for letters over symbols and a preference for more frequently used letters. This orthographic tuning is also the hallmark of the type II component. However, while the amplitude of this component increases with increasing visual complexity through the addition of more symbols or letters to a stimulus (*Tarkiainen et al., 1999*), an important aspect of the type II is that manipulations that reduce readability, such as the addition of visual noise, decrease its amplitude, even though the overall visual complexity of the stimulus is increased. It is in this aspect that the convolution units in our models are not necessarily a good fit for the type II response, since their tight receptive field strongly ties their activation to that of their counterparts in the previous layer. Increased activation in the early convolution layers tends to carry over, with the later convolution layers having only limited capabilities for filtering out noise. With small vocabularies, we did observe noise-filtering behavior in the fifth convolution layer, but less so when vocabulary size was increased from 250 to 1k and the effect was gone with a vocabulary of 10k.

In the model, convolution units are followed by pooling units, which serve the purpose of stratifying the response across changes in position, size, and rotation within the receptive field of the pooling unit. Hence, the effect of small differences in letter shape, such as the usage of different fonts, was only present in the early convolution layers, in line with findings in the EEG literature (*Chauncey et al., 2008*; *Grainger and Holcomb, 2009*; *Hauk and Pulvermüller, 2004*). However, the ability of pooling

units to stratify such differences depends on the size of their receptive field, which is determined by the number of convolution-and-pooling layers. As a consequence, the response profiles of the subsequent fully connected layers was also very sensitive to the number of convolution-and-pooling layers. The optimal number of such layers is likely dependent on the input size and pooling strategy. Given the VGG-11 design of doubling the receptive field after each layer, combined with an input size of $225 \times 225$ pixels, the optimal number of convolution-and-pooling layers for our model was five, or the model would struggle to produce response profiles mimicking those of the type II component in the subsequent fully connected layers (*Figure 4*).

The fully connected layers pool information across the entire input image. Their large receptive fields make them more effective in isolating orthographic information from noise than the convolution layers (*Figure 3*) and their response profile matches that of the type II component well, also when the model has a large vocabulary. A minimum of two fully connected layers was needed to achieve this in our case, and adding more fully connected layers would make them behave more like the N400m component (*Figure 4*). Moreover, units in the fully connected layers can see multiple letters, making them sensitive not only to letter frequency, but also bigram frequency, allowing them to replicate these effects on the type II component (*Woolnough et al., 2021*).

The fully connected linear layers in the model show a negative correlation with font size. While the N400 has been shown to be unaffected by font size during repetition priming (*Chauncey et al., 2008*), it has been shown that in the absence of priming, larger font sizes decrease the evoked activity in the 300–500 ms window (*Bayer et al., 2012*; *Schindler et al., 2018*). Those studies refer to the activity within this time window, which seems to encompass the N400, as early posterior negativity (EPN). What possibly happens in the model is that an increase in font size causes an initial stronger activation in the first layers, due to more convolution units receiving input. This leads to a better signal-to-noise ratio (SNR) later on, as the noise added to the activation of the units remains constant while the amplitude of the input signal increases. A better SNR translates ultimately in less co-activation of units corresponding to orthographic neighbors in the final layers, hence to a decrease in overall layer activity.

The final fully connected layer, the output layer, represents the model's notion of a mental lexicon: a collection of known word-forms, not yet linked to their semantic meaning (which we did not model at all). Since orthographic word-form and semantic meaning are generally only very loosely coupled (*Monaghan et al., 2014*), the lexicon often appears as a separate unit in computational models (*Woollams, 2015*). At face value, since we employed cross-entropy as loss function during training, the output layer simulates the lexicon following a localist structural approach, which harks back to the earliest models of visual word recognition (*Morton, 1969*; *McClelland and Rumelhart, 1981*). In the localist approach, each word is represented by a dedicated unit that should only be active when the sensory version of that word (in this study, a bitmap image containing the word as rendered text) is provided as input to the model, to the exclusion of all other words in the vocabulary. This is known in the machine learning literature as 'one-hot' encoding. The localist approach is often contrasted with the distributed approach, wherein each word is represented by a unique combination of multiple output units that respond to subword parts (e.g., n-grams, pairings of letters in the first and final position, etc.), which is a more efficient encoding scheme (*Seidenberg and McClelland, 1989*). Both the localist and distributed approaches have their strengths and weaknesses when it comes to explaining experimental results, and the debate on how exactly the human lexicon is encoded in the brain is far from settled (*Woollams, 2015*). Interestingly, because our model covers the entire process from pixel to word-form, its simulation of the lexicon is not strictly localist, but combines both localist and distributed approaches.

When the input image contains a word that is present in the model's vocabulary, it is true that the corresponding word-specific unit in the output layer will have the highest amount of activation of all units in the output layer. However, given the connectionist nature of our model, all units tuned to orthographically similar words will co-activate to some degree. This inaccuracy is important when it comes to simulating the amplitude of the N400m component in response to pseudowords, which is as large as (and sometimes even larger than) the response to real words. In the model, a pseudoword does not have a dedicated output unit, but given a large enough vocabulary, there will be enough units that respond to orthographically similar words to produce a large amount of activity nonetheless. Crucially, letter strings that are not word-like, such as consonant strings, will not activate many units

in the output layer, which is an important difference between the response profiles of the N400m and the type II. Furthermore, in our final model, the first two fully connected layers have fewer units (4096) than the number of words in the vocabulary (10,000), which produces a distributed representation, where one unit is involved in the encoding of many words. The response profiles of both the second fully connected layer (distributed representation) and the output layer (localist representation) resemble that of the N400m response. Hence, our model does not provide direct cause for preferring a localist or distributed approach when modeling the lexicon, but shows how both approaches interact when creating a model of the entire process of sensory input to lexicon: the localist approach can function as training objective and a distributed representation emerges in the hidden layers.

## On the importance of experimental contrasts and qualitative analysis of the model

When the same stimuli can be presented to both human participants and model, model–brain correspondence can be evaluated quantitatively through metrics such as correlation, linear regression (brainscore), and RSA. Although stimuli need not be designed to fall into a priori defined categories, it still pays off to have experimental contrasts in the experiment. For one, quantitative metrics are more meaningful when the variation in the data is due to multiple causes and no single cause (e.g., word length or visual complexity) dominates the variation. More importantly, as demonstrated by our results, quantitative metrics alone can give a distorted view of model–brain correspondence. For example, it is clear that the amount of visual complexity in a stimulus (e.g., visual noise, number of letters, font size) accounts for a large portion of the variation in the evoked responses (*Figure 1D*) as well as all layers of the model (*Figure 7*). Since even a CNN with randomly initialized weights will mimic the variation due to visual complexity (*Figure 3—figure supplement 2*, top row), one can expect a high correlation between model and brain whenever there are large differences in visual complexity between stimuli. In our case, the inclusion of stimuli that occur less frequently in our natural environment, such as symbol strings, consonant strings and pseudowords, matched in length with real words, were very informative causes of signal-variation beyond visual complexity, and were essential for the evaluation of the models' simulation of the neurofunctionally relevant type II and N400m components. The contrast between noise-embedded words and noise-free stimuli sometimes caused a negative correlation between model and brain, which is a clear sign that the model was a poor fit. One should keep this in mind when using quantitative metrics that are insensitive to the directionality of the correlation, as they can hide such a clear deficit (*Figure 6—figure supplement 1*).

Overall, we found that a qualitative evaluation of the response profiles was more helpful than correlation scores. Often, a deficit in the response profile of a layer that would cause a decrease in correlation on one condition would be masked by an increased correlation in another condition. A notable example is the necessity for frequency balancing the training data when building models with a vocabulary of 10,000. Going by correlation score alone, there does not seem to be much difference between the model trained with and without frequency balancing (*Figure 3A*, fifth row versus bottom row). However, without frequency balancing, we found that the model did not show a response profile where consonant strings were distinguished from words and pseudowords (*Figure 3A*, fifth row), which is an important behavioral trait that sets the N400m component apart from the type II component (*Figure 1D*). This underlines the importance of the qualitative evaluation in this study, which was only possible because of a straightforward link between the activity simulated within a model to measurements obtained from the brain, combined with the presence of clear experimental conditions.

Since evoked activity is produced by large amounts of neurons firing at the same time (*Hämäläinen et al., 1993*; *Murakami and Okada, 2006*), we chose in this study to use the total amount of activity within the layers of the model as direct simulations of the amplitudes of the various components of the MEG response. This simulation could perhaps be improved through the use of more biologically realistic methods, such as neural mass models (*David et al., 2005*) and dynamic causal modeling (*Friston et al., 2003*).

## Limitations of the current model and the path forward

The VGG-11 architecture was originally designed to achieve high image classification accuracy on the ImageNet challenge (*Simonyan and Zisserman, 2015*). Although we have introduced some

modifications that make the model more biologically plausible, the final model is still incomplete in many ways as a complete model of brain function during reading.

An important limitation of the model is the lack of feed-back activity, which is an important hall-mark of the reading process and is likely to already have an influence on early brain activity in visual cortex (*Heilbron et al., 2020*). For one, this means that our model is not able to capture the effect of the previous stimulus on the current one, such as priming effects (*Grainger and Holcomb, 2009*; *Kutas and Federmeier, 2011*). But even when considering stimuli in isolation, there are feed-back effects that the model cannot simulate. While the design process of the model has been guided by the experimental contrasts of *Vartiainen et al., 2011*, we further evaluated the final modal on several more contrasts (*Figure 7*). For some contrasts, it did not succeed in reproducing the experimental effects as typically found in EEG/MEG studies. The fact that our feed-forward model failed to simulate the effects of word frequency on the N400m, even after frequency balancing of the training data, suggests that this effect may be largely driven by feed-back activity, as for example modeled by *Nour Eddine et al., 2024*.

Another limitation of the current model is the lack of an explicit mapping from the units inside its layers to specific locations in the brain at specific times. The temporal ordering of the components is simulated correctly, with the response profile matching that of the type I occurring in layers that precede those matching the type II, followed by the layers corresponding to the N400m. Further-more, every component is best modeled by a different type of layer, with the type I best described by convolution-and-pooling, the type II by fully connected linear layers, and the N400m by a one-hot encoded layer. However, there is no clear relationship between the number of layers the signal needs to traverse in the model to the processing time in the brain. Even if one considers that the operations performed by the initial two convolution layers happen in the retina rather than the brain, the signal needs to propagate through three more convolution layers to reach the point where it matches the type II component at 140–200 ms, but only through one more additional layer to reach the point where it starts to match the N400m component at 300–500 ms. Still, cutting down on the number of times convolution is performed in the model seems to make it unable to achieve the desired suppres-sion of noise (*Figure 4*). This raises the question what the brain is doing during the time between the type II and N400m component that seems to take so long. It is possible that the timings of the MEG components are not indicative solely of when the feed-forward signal first reaches a certain location, but are rather dictated by the resolution of feed-forward and feed-back signals (*Nour Eddine et al., 2024*).

In this paper we have restricted our simulations to feed-forward processes. Now, the way is open to incorporate convolution-and-pooling principles in models of reading that simulate feed-back processes as well, which should allow the model to capture more nuance in the type II and N400m components, as well as extend the simulation to encompass a realistic semantic representation. A promising way forward may be to use a network architecture like cornet (*Kubilius et al., 2019*) that performs convolution multiple times in a recurrent fashion, yet simultaneously propagates activity forward after each pass. The introduction of recursion into the model will furthermore align it better with traditional-style models, since it can cause a model to exhibit attractor behavior (*McLeod et al., 2000*), which will be especially important when extending the model into the semantic domain. Furthermore, convolution-and-pooling has recently been explored in the domain of predic-tive coding models (*Ororbia and Mali, 2023*), a type of model that seems particularly well suited to model feed-back processes during reading (*Gagl et al., 2020*; *Heilbron et al., 2020*; *Nour Eddine et al., 2024*).

Despite its limitations, our model is an important milestone for computational models of reading that leverages deep learning techniques to encompass the entire computational process starting from raw pixels values to representations of word-forms in the mental lexicon. The overall goal is to work towards models that can reproduce the dynamics observed in brain activity observed in the large number of neuroimaging experiments with human volunteers that have been performed over the last few decades. To achieve this, models need to be able to operate on more realistic inputs than a collection of predefined lines or letter banks (*McClelland and Rumelhart, 1981*; *Coltheart et al., 2001*; *Laszlo and Armstrong, 2014*; *Heilbron et al., 2020*; *Nour Eddine et al., 2024*). We have shown that even without feed-back connections, a CNN can simulate the behavior of three important MEG evoked components across a range of experimental conditions, but only if unit activations are

noisy and the frequency of occurrence of words in the training dataset mimics their frequency of use in actual language.

## Materials and methods
### MEG study design and data analysis

The MEG data that had been collected by *Vartiainen et al., 2011* was re-analyzed for the current study using MNE-python (*Gramfort et al., 2013*) and FreeSurfer (*Dale et al., 1999*). The study received ethical approval from the Ethics Committee of the Helsinki and Uusimaa Hospital District and informed written consent, and consent to publish, was obtained from each participant. We refer to the original publication for additional details on the study protocol and data collection process.

Simultaneous MEG and EEG was recorded from 15 Finnish participants (7 females; age 20–49 years, mean 27 years, all included in the analysis). The stimuli consisted of visually presented Finnish words, pseudowords (pronounceable but meaningless), consonant strings (random consonants), symbol strings (randomly formed from 10 possible shapes), and words embedded in high-frequency visual noise (*Figure 1A*). Each stimulus category contained 112 different stimuli and each stimulus contained 7–8 letters or symbols. The stimuli were presented sequentially in a block design. Each block started with a random period of rest (0–600 ms), followed by seven stimuli of the same category. Each stimulus was shown for 300 ms with an inter-stimulus interval of 1500 ms of gray background. In addition to the five stimulus conditions (16 blocks each), there were 16 blocks of rest (12 s). An additional five target blocks were added, in which one stimulus appeared twice in a row, that the participants were asked to detect and respond to with a button press. The target blocks were not included in the analysis. The same paradigm was applied in an fMRI–EEG recording of the same participants as well (data not used here).

The data were recorded with a Vector View device (Elekta Neuromag, now MEGIN Oy, Finland) in a magnetically shielded room. In addition, vertical and horizontal electro-oculography (EOG) were recorded. All MEG data analysis was performed using the MNE-Python software (*Gramfort et al., 2013*). The signals were bandpass filtered at 0.03–200 Hz and digitized at 600 Hz. For analysis, the data were further processed using the maxfilter software (MEGIN Oy, Finland) and bandpass filtered at 0.1–40 Hz. Eye-movement and heart-beat artifacts were reduced using independant component analysis (ICA): the signal was high-passed at 1 Hz, and components with correlations larger than 5 standard deviations with any of the EOG channels or an estimated electro-cardiogram (based on the magnetometers without the maxfilter applied) were removed (3–7 out of 86–108 components). Epochs were cut −200 to 1000 ms relative to the onset of the stimulus and baseline-corrected using the pre-stimulus interval. Participant-specific noise covariance matrices of the MEG sensors were estimated using the pre-stimulus interval.

Source estimation of the sensor-level signals was performed using two methods. The first, MNE-dSPM (*Dale et al., 2000*), yields a comprehensive map of where activity can be found, which is used in this study mostly to provide context for a deeper investigation into three peaks of activity that can be observed along the ventral stream. The second, guided ECD modeling (*Salmelin, 2010*), summarizes the high-dimensional data as a sparse set of ECDs, each one capturing an isolated spatiotemporal component, allowing us to study selected components in more detail. The advantage of this sparse method over distributed source estimation techniques such as the MNE or beamformers, is that in cases like *Vartiainen et al., 2011*, a single ECD may conveniently capture an isolated component of the evoked response. Furthermore, this sparsity makes it straightforward to take individual differences regarding the cortical origin of the evoked component into account, since the location and orientation of the ECDs were fitted to optimally explain the data of each individual. For each individual, the matching between ECDs and the type I, type II, and N400m responses was performed based on their approximate location, timing of peak activity, and behavior across the different stimulus types. The groups do not necessarily contain an ECD for each of the 15 participants, as some participants did not have an ECD to contribute that matched well enough in position and/or timing. The number of participants contributing an ECD to each group were: type I: 14, type II: 14, and N400m: 15.

For both ECD and MNE-dSPM source estimation, three-layer boundary element method (BEM) forward models were used. These models were based on T1-weighted anatomical images that were acquired using a 3T Signa excite MRI scanner. FreeSurfer (*Dale et al., 1999*) was used to obtain the

surface reconstructions required to compute the BEM models. *Vartiainen et al., 2011* defines three groups of ECDs in the left-hemisphere, associated with the type I, type II, and N400m responses. The locations and orientations of the ECDs in these groups were re-used in the current study. For each individual epoch, the signal at each of the three ECDs and a cortex-wide MNE-dSPM source estimate was computed using the noise covariance matrices and BEM forward models. Finally, the locations of the ECDs and MNE-dSPM source points were morphed to FreeSurfer's template brain.

## Computational models

The models of the computational process underlying the brain activity observed during the MEG experiment used a VGG-11 (*Szegedy et al., 2015*) network architecture, pretrained on ImageNet (*Russakovsky et al., 2015*), as defined in the TorchVision (*Marcel and Rodriguez, 2010*) module of the PyTorch (*Paszke et al., 2019*) package. This architecture consists of five convolution layers (with the final three performing convolution twice), followed by two fully connected linear layers, terminating in a fully connected linear output layer (*Figure 2A*). Each individual convolution step is followed by batch normalization (*Ioffe and Szegedy, 2015*) and max-pooling. Every unit, including the output units, used the following non-linear activation function:

$$\text{noisyReLU}(x) = \max(x + N(0, \sigma_{\text{noise}}), 0), \tag{1}$$

where $N$ denotes Gaussian distributed random noise with zero mean and standard deviation $\sigma_{\text{noise}}$. For version of the model with noisy activations $\sigma_{\text{noise}} = 0.1$, and for versions without $\sigma_{\text{noise}} = 0$.

The model was trained to perform visual word recognition using a training set of approximately 1,000,000 images, where each image depicted a word, rendered in varying fonts, sizes, and rotations (*Figure 2B*). An independent test set containing 100,000 images was used to track the performance of the model during training. None of the images in the training or test set were an exact duplicate of the images used as stimuli in the MEG experiment. Therefore, the stimulus set can be treated as an independent validation set. The task for the model was to identify the correct word (regardless of the font, size, and rotation used to render the text) by setting the corresponding unit in the output layer to a high value.

Different versions of the model were created using vocabulary sizes of 250, 1000, and 10,000 Finnish words. The vocabulary was compiled using the 112 words used in the MEG, extended with the most frequently used Finnish words, as determined by *Kanerva et al., 2014*, having a length of 3–9, excluding proper names and using their lemma form. No symbol strings, consonant strings, nor pseudowords were present in the training set.

For the training sets that do not take word frequency into account, each word was rendered 100 times in various fonts, sizes, and rotations. When word frequency was taken into account, the number of times each word occurred in the training set c was scaled by its frequency $f$ in a large Finnish text corpus (*Kanerva et al., 2014*) in the following manner:

$$r = \left[\frac{f}{f_0}\right]^s, \tag{2}$$

$$c = \left\lfloor N\frac{r}{\sum \mathbf{r}} \right\rfloor, \tag{3}$$

where $f_0$ is the frequency of the least commonly occurring word in the training vocabulary, $0 \le s \le 1$ is the amount of frequency balancing to apply, $r$ is the frequency ratio, $\sum \mathbf{r}$ is the total frequency ratio of all words in the training vocabulary, and $N$ is the desired number of items in the training set, in our case 1,000,000. The actual number of items in the training set will deviate a little from $N$, as the floor operation causes $\sum \mathbf{c} \approx N$. The number of times a word occurred in the test set was always 10.

Images were assembled by overlaying the rendered text onto an image of visual noise. Visual noise was rendered by randomly selecting a gray-scale value (10–100%) for each pixel. Next, if the training image contained text, the word was rendered in uppercase letters, using a randomly chosen font (15 possible fonts), font size (14 pt to 32 pt), and rotation (–20° to 20°). The list of fonts consisted of: Ubuntu Mono, Courier, Luxi Mono Regular, Lucida Console, Lekton, Dejavu Sans Mono, Times New Roman, Arial, Arial Black, Verdana, Comic Sans MS, Georgia, Liberation Serif, Impact, and Roboto Condensed.

For every variation of the model, the training procedure was the same. The training objective was to minimize the cross-entropy loss between the model output and a one-hot encoded vector of a length equal to the vocabulary size, indicating the target word. Optimization of the model's parameters was performed using stochastic gradient descend with an initial learning rate of 0.01, which was reduced to 0.001 after 10 epochs (an 'epoch' in this context refers to one sequence of all 1,000,000 training images), and a momentum of 0.9. During training, the noisy-ReLU activation function was temporarily disabled for the output units, and re-activated during evaluation and normal operation of the model. In total, training was performed for 20 epochs, by which point the performance on the test set, for all variations of the model, had plateaued at around 99.6%.

## Comparison between model and MEG data

To compare the model with the MEG responses, statistical summaries were made of the high-dimensional data, that we refer to as 'response profiles', that could be compared either qualitatively by examining the pattern of activation across stimulus types, and quantitatively by computing Pearson correlation between them.

To construct the response profiles for a model, it was applied to the bitmap images that were used during the MEG experiment (padded to be $256 \times 256$ pixels with 50% gray-scale pixels). The activity at each layer of the model was quantified by recording the ReLU activation right before the maxpool operation, and computing the $\ell_2$ norm across the units. This yielded, for each stimulus image, a single number for each layer in the model representing the amount of activity within that layer. For each layer, the response profile was z-scored across the stimulus images.

To construct the response profiles for the type I, type II, and N400m components, the neural activity represented by the corresponding ECD was quantified by taking the mean of its timecourse during an appropriate time window (indicated in *Figure 1C*, 64–115 ms for the type I, 140–200 ms for the type II, and 300–400 ms for the N400m, after stimulus onset). This yielded for each stimulus image a single number that indicates the amount of signal at the ECD. Collected across all stimuli, these numbers form the response pattern of each ECD. The response patterns were first z-scored across the stimulus images independently for each participant and later averaged across participants for the grand-average analysis.

The MNE-dSPM source estimates used 4000 source points, distributed evenly across the cortex. The response profile at each source point was computed by taking the maximum signal value of the source point's timecourse within the same time window as the ECD analysis. This yielded for each stimulus image, a single number for each source point for each participant, indicating the amount of signal at the source point. The response patterns were first z-scored across the stimulus images independently for each participant and later averaged across participants during the grand-average analysis.

## Statistics

The comparisons between response profiles obtained from the models and the MEG data were performed through Pearson correlation (*Figures 3B, 4B, 6, and 7*). In cases where distributions were tested for a significant difference in mean (*Figures 3A and 4A*), this was done through two-tailed paired *t*-tests, as implemented in SciPy (*Virtanen et al., 2020*). In *Figure 1D*, multiple participants contribute to each data point in the distributions, hence a linear mixed effects model was used to compare distributions, with participants modeled as random effect (both slope and intercept), as implemented in the Julia MixedModels package (*Bates et al., 2023*).

The noise ceilings, indicating the maximum obtainable correlation between the response profile of a model and those of the evoked components (*Figures 3B and 4B*), were estimated following a method proposed by *Lage-Castellanos et al., 2019*, which is based on the between-participant variation:

$$n = \frac{\sqrt{\sigma_\mu^2 - \frac{\mu_{\sigma^2}}{N}}}{\sigma_\mu}, \tag{4}$$

where $\sigma_\mu$ is the standard deviation (across the stimuli) of the grand-average response profile, $\mu_{\sigma^2}$ is the mean (across the stimuli) variation across the participant-specific response profiles, and $N$ is the number of participants.

## Post hoc experimental contrasts

The final model was not only tested with the stimuli of *Vartiainen et al., 2011*, but also sets of stimuli designed with additional experimental contrasts (*Figure 7*). For each contrast, a different set of stimuli was generated. Unless mentioned otherwise, stimuli were text strings rendered on a gray background in the Arial font at a fontsize of 23 pixels.

### Pure noise

1000 images were generated containing a gray background, which was blended with Gaussian noise with varying levels between 0% and 100%, with no rendered text.

### Noisy word

All words of length 8 were selected from the original 10k vocabulary and rendered. They were blended with Gaussian noise at random levels between 0% and 50%.

### Num. letters

All words in the 10k vocabulary were rendered.

### Num. symbols

For each word in the 10k vocabulary, a symbol string of matching length was generated using the unicode characters 'square', 'circle', 'triangle up', 'triangle down', 'diamond', and rendered.

### Word versus symbols

The stimulus lists used for the 'Num. letters' and 'Num. symbols' contrasts were combined. For this contrast, the model response profile was correlated with a vector containing zeros indicating symbols and ones indicating words.

### Font family

All words in the 10k vocabulary were rendered in both the Impact and Comic Sans fonts. For this contrast, the model response profile was correlated with a vector containing zeros indicating stimuli rendered in the Impact font, and ones indicating stimuli rendered in the Comic Sans font.

### Font size

All words of length 8 were selected from the original 10k vocabulary. Each selected word was rendered at 8 fontsizes ranging from 14 to 32 pixels.

### Letter freq

10,000 random letter strings of length 8 were generated and rendered. In this contrast, the model response profile was correlated with the total letter frequency of each random letter string in the 10k vocabulary.

### Bigram freq

10,000 strings consisting of 4 bigrams were generated and rendered. Since there is generally a strong correlation between letter and bigram frequency, the random bigrams were selected in a fashion designed to minimize this confounding factor. Bigrams were randomly selected from a set of 50 common bigrams (freq ≥ 30,000) with the lowest letter frequency, and a set of 50 uncommon bigrams (freq < 10,000) with the highest letter frequency. Bigram and letter frequencies were computed on the 10k vocabulary.

## Word freq

All words in the 10k vocabulary were rendered. The word frequency was computed as the square root of the number of times the word occured in a large corpus of Finnish text (*Kanerva et al., 2014*).

## Orth. neighb

All words in the 10k vocabulary were rendered. The orthographic neighborhood size was computed as the number of words in the 10k vocabulary with a Levenshtein distance of 1 to the target word.

## Acknowledgements

We acknowledge the computational resources provided by the Aalto Science-IT project. The funders had no role in study design, data collection, and analysis, decision to publish, or preparation of the manuscript.

## Additional information

### Funding

| Funder | Grant reference number | Author |
|---|---|---|
| Research Council of Finland | #310988 | Marijn van Vliet |
| Research Council of Finland | #315553 | Riitta Salmelin |
| Research Council of Finland | #343385 | Marijn van Vliet |
| Research Council of Finland | #355407 | Riitta Salmelin |
| Sigrid Juséliuksen Säätiö | | Riitta Salmelin |

The funders had no role in study design, data collection, and interpretation, or the decision to submit the work for publication.

### Author contributions

Marijn van Vliet, Conceptualization, Data curation, Software, Formal analysis, Funding acquisition, Investigation, Visualization, Methodology, Writing – original draft, Writing – review and editing; Oona Rinkinen, Formal analysis, Investigation; Takao Shimizu, Software, Investigation, Visualization; Anni-Mari Niskanen, Investigation, Visualization; Barry Devereux, Riitta Salmelin, Conceptualization, Supervision, Writing – review and editing

### Author ORCIDs

Marijn van Vliet https://orcid.org/0000-0002-6537-6899
Riitta Salmelin https://orcid.org/0000-0003-2499-193X

### Ethics

The MEG data that had been collected by Vartiainen et al. (2011) was re-analyzed for the current study. The original study received ethical approval from the Ethics Committee of the Helsinki and Uusimaa Hospital District and informed written consent, and consent to publish, was obtained from each participant.

Reviewer #2 (Public review): https://doi.org/10.7554/eLife.96217.3.sa1
Reviewer #3 (Public review): https://doi.org/10.7554/eLife.96217.3.sa2
Author response https://doi.org/10.7554/eLife.96217.3.sa3

## Additional files

### Supplementary files
MDAR checklist

### Data availability
All data and materials created during this study are publically available. The programming code for training the models and reproducing the results of this study and generating all the figures is available at https://github.com/wmvanvliet/viswordrec-baseline (copy archived at *van Vliet, 2025*; license: BSD-3-clause). A public data package containing all models, training sets, and MEG derivatives necessary for performing the model–brain comparison and is available at https://osf.io/nu2ep. For the comparison between model and brain, MEG data was re-used from a previous study (*Vartiainen et al., 2011*) and the informed consent obtained from the participants in that study prohibits sharing of personal data, even after removal of direct identifiers, hence only MEG data that has been aggregated across participants is made publically available. This does not have a big impact regarding the reproducibility of the present study, as the only parts relying on an individual's data are the computation of the noise ceilings, the results of which are included in the public data package, and the locations of the dipoles on the cortex, which are solely used for illustrative purposes in *Figure 1*. The reuse of the personal data collected in *Vartiainen et al., 2011* for other research purposes requires a new ethical review, and such requests should be addressed to Riitta Salmelin (riitta.salmelin@aalto.fi).

The following dataset was generated:

| Author(s) | Year | Dataset title | Dataset URL | Database and Identifier |
|---|---|---|---|---|
| van Vliet M | 2022 | Convolutional networks can model the functional modulation of MEG responses during reading | https://doi.org/10.17605/OSF.IO/NU2EP | Open Science Framework, 10.17605/OSF.IO/NU2EP |

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
