## [Editor Report · eLife Assessment]

van Vliet and colleagues show a **useful** correlation between internal states of a convolutional neural network (CNN) trained on visual word stimuli with three specific components of evoked MEG potentials during reading in humans. The findings are **solid**, though quantitative evidence that model can produce any of the phenomena that the human visual system is known to have (e.g., feedback connections, sensitivity to word frequency), or that it has comparable performance to human behaviour (i.e., similar task accuracy with a comparable pattern of mistakes) would make the conclusions much stronger.

---

## [Referee Report · Reviewer #2 (Public review)]

van Vliet and colleagues present results of a study correlating internal states of a convolutional neural network trained on visual word stimuli with evoked MEG potentials during reading.

In this study, a standard deep learning image recognition model (VGG-11) trained on a large natural image set (ImageNet) that begins illiterate but is then further trained on visual word stimuli, is used on a set of predefined stimulus images to extract strings of characters from "noisy" words, pseudowords and real words. This methodology is used in hopes of creating a model which learns to apply the same nonlinear transforms that could be happening in different regions of the brain - which would be validated by studying the correlations between the weights of this model and neural responses. Specifically, the aim is that the model learns some vector embedding space, as quantified by the spread of activations across a layer's weights (L2 Norm prior to ReLu Activation Function), for the different kinds of stimuli, that creates a parameterized decision boundary that is similar to amplitude changes at different times for a MEG signal. More importantly, the way that the stimuli are ordered or ranked in that space should be separable to the degree we see separation in neural activity. This study does show that the layer weights corresponding to five different broad classes of stimuli do statistically correlate with three specific components in the ERP. However, I believe there are fundamental theoretical issues that limit the implications of the results of this study.

As has been shown over many decades, there are many potential computational algorithms, with varied model architectures, that can perform the task of text recognition from an image. However, there is no evidence presented here that this particular algorithm has comparable performance to human behavior (i.e. similar accuracy with a comparable pattern of mistakes). This is a fundamental prerequisite before attempting to meaningfully correlate these layer activations to human neural activations. Therefore, it is unlikely that correlating these derived layer weights to neural activity provides meaningful novel insights into neural computation beyond what is seen using traditional experimental methods.

One example of a substantial discrepancy between this model and neural activations is that, while incorporating frequency weighting into the training data is shown to slightly increase neural correlation with the model, Figure 7 shows that no layer of the model appears directly sensitive to word frequency. This is in stark contrast to the strong neural sensitivity to word frequency seen in EEG (e.g. Dambacher et al 2006 Brain Research), fMRI (e.g. Kronbichler et al 2004 NeuroImage), MEG (e.g. Huizeling et al 2021 Neurobio. Lang.), and intracranial (e.g. Woolnough et al 2022 J. Neurosci.) recordings. Figure 7 also demonstrates that late stages of the model show a strong negative correlation with font size, whereas later stages of neural visual word processing are typically insensitive to differences in visual features, instead showing sensitivity to lexical factors.

Another example of the mismatch between this model and visual cortex is the lack of feedback connections in the model. Within visual cortex there are extensive feedback connections, with later processing stages providing recursive feedback to earlier stages. This is especially evident in reading, where feedback from lexical level processes feeds back to letter level processes (e.g. Heilbron et al 2020 Nature Comms.). This feedback is especially relevant for reading of words in noisy conditions, as tested in the current manuscript, as lexical knowledge enhances letter representation in visual cortex (the word superiority effect). This results in neural activity in multiple cortical areas varying over time, changing selectivity within a region at different measured time points (e.g. Woolnough et al 2021 Nature Human Behav.), which in the current study is simplified down to three discrete time windows, each attributed to different spatial locations.

The presented model needs substantial further development to be able to replicate, both behaviorally and neurally, many of the well-characterized phenomena seen in human behavior and neural recordings that are fundamental hallmarks of human visual word processing. Until that point it is unclear what novel contributions can be gleaned from correlating low dimensional model weights from these computational models with human neural data.

The revised version of this manuscript has not addressed these concerns.

---

## [Referee Report · Reviewer #3 (Public review)]

Summary:

The authors investigate the extent to which the responses of different layers of a vision model (VGG-11) can be linked to the cascade of responses (namely, type-I, type-II and N400) in the human brain when reading words. To achieve maximal consistency between, they add noisy-activations to VGG and finetune it on a character recognition task. In this setup, they observe various similarities between the behavior of VGG and the brain when presented with various transformations of the words (added noise, font modification etc).

Strengths:

- The paper is well written and well presented

- The topic studied is interesting.

- The fact that the response of the CNN on unseen experimental contrasts such as adding noise correlated with previous results on the brain is compelling.

Weaknesses:

- The paper is rather qualitative in nature. In particular, the authors show that some resemblance exists between the behavior of some layers and some parts of the brain, but it is hard to quantitively understand how strong the resemblences are in each layer, and the exact impact of experimental settings such as the frequency balancing (which seems to only have a very moderate effect according to figure 5)

- The experiments only consider a rather outdated vision model (VGG)

Comments on revisions:

After rebuttal, the authors significantly strengthened their results. I now find the paper much more convincing, and thank the author for their careful consideration of the reviewers' suggestions.

---

## [Author Response]

The following is the authors’ response to the original reviews.

We thank the reviewers for their efforts. They have pointed out several shortcomings and made very helpful suggestions. Based on their feedback, we have substantially revised the manuscript and feel the paper has been much improved because of it.

Notable changes are:

(1) As our model does not contain feed-back connections, the focus of the study is now more clearly communicated to be on feed-forward processes only, with appropriate justifications for this choice added to the Introduction and Discussion sections. Accordingly, the title has been changed to include the term “feed-forward”.

(2) The old Figure 5 has been removed in favor of reporting correlation scores to the right of the response profiles in other figures.

(3) We now discuss changes to the network architecture (new Figure 5) and fine-tuning of the hyperparameters (new Figure 6) in the main text instead of only the Supplementary Information.

(4) The discussion on qualitative versus quantitative analysis has been extended and given its own subsection entitled “On the importance of experimental contrasts and qualitative analysis of the model”.

Below, we address each point that the reviewers brought up in detail and outline what improvements we have made in the revision to address them.

**Reviewer #1 (Public Review):**
Summary:This study trained a CNN for visual word classification and supported a model that can explain key functional effects of the evoked MEG response during visual word recognition, providing an explicit computational account from detection and segmentation of letter shapes to final word-form identification.Strengths:This paper not only bridges an important gap in modeling visual word recognition, by establishing a direct link between computational processes and key findings in experimental neuroimaging studies, but also provides some conditions to enhance biological realism.Weaknesses:The interpretation of CNN results, especially the number of layers in the final model and its relationship with the processing of visual words in the human brain, needs to be further strengthened.

We have experimented with the number of layers and the number of units in each layer. In the previous version of the manuscript, these results could be found in the supplementary information. For the revised version, we have brought some of these results into the main text and discuss them more thoroughly.

We have added a figure (Figure 5 in the revised manuscript) showing the impact of the number of convolution and fully-connected layers on the response profiles of the layers, as well as the correlation with the three MEG components.

We discuss the figure in the Results section as follows:

“Various variations in model architecture and training procedure were evaluated. We found that the number of layers had a large impact on the response patterns produced by the model (Figure 5). The original VGG-11 architecture defines 5 convolution layers and 3 fully connected layers (including the output layer). Removing a convolution layer (Figure 5, top row), or removing one of the fully connected layers (Figure 5, second row), resulted in a model that did exhibit an enlarged response to noisy stimuli in the early layers that mimics the Type-I response. However, such models failed to show a sufficiently diminished response to noisy stimuli in the later layers, hence failing to produce responses that mimic the Type-II or N400m, a failure which also showed as low correlation scores.

Adding an additional convolution layer (Figure 5, third row) resulted in a model where none of the layer response profiles mimics that of the Type-II response. The Type-II response is characterized by a reduced response to both noise and symbols, but an equally large response to consonant strings, real and pseudo words. However, in the model with an additional convolution layer, the consonant strings evoked a reduced response already in the first fully connected layer, which is a feature of the N400m rather than the Type-II. These kind of subtleties in the response pattern, which are important for the qualitative analysis, generally did not show quantitatively in the correlation scores, as the fully connected layers in this model correlate as well with the Type-II response as models that did show a response pattern that mimics the Type-II.

Adding an additional fully connected layer (Figure 5, fourth row) resulted in a model with similar response profiles and correlation with the MEG components as the original VGG-11 architecture (Figure 5, bottom row) The N400m-like response profile is now observed in the third fully connected layer rather than the output layer. However, the decrease in response to consonant strings versus real and pseudo words, which is typical of the N400m, is less distinct than in the original VGG-11 architecture.”

And in the Discussion section:

“In the model, convolution units are followed by pooling units, which serve the purpose of stratifying the response across changes in position, size and rotation within the receptive field of the pooling unit. Hence, the effect of small differences in letter shape, such as the usage of different fonts, was only present in the early convolution layers, in line with findings in the EEG literature (Chauncey et al., 2008; Grainger & Holcomb, 2009; Hauk & Pulvermüller, 2004). However, the ability of pooling units to stratify such differences depends on the size of their receptive field, which is determined by the number of convolution-and-pooling layers. As a consequence, the response profiles of the subsequent fully connected layers was also very sensitive to the number of convolution-and-pooling layers. The optimal number of such layers is likely dependent on the input size and pooling strategy. Given the VGG-11 design of doubling the receptive field after each layer, combined with an input size of 225×225 pixels, the optimal number of convolution-andpooling layers for our model was five, or the model would struggle to produce response profiles mimicking those of the Type-II component in the subsequent fully connected layers (Figure 5).”

**Reviewer #1 (Recommendations For The Authors):**
(1) The similarity between CNNs and human MEG responses, including type-I (100ms), type-II (150ms), and N400 (400ms) components, looks like separately, lacking the sequential properties among these three components. Is the recurrent neural network (RNN), which can be trained to process and convert a sequential data input into a specific sequential data output, a better choice?

When modeling sequential effects, meaning that the processing of the current word is influenced by the word that came before it, such as priming and top-down modulations, we agree that such a model would indeed require recurrency in its architecture. However, we feel that the focus of modeling efforts in reading has been overwhelmingly on the N400 and such priming effects, usually skipping over the pixel-to-letter process. So, for this paper, we were keen on exploring more basic effects such as noise and symbols versus letters on the type-I and type-II responses. And for these effects, a feed-forward model turns out to be sufficient, so we can keep the focus of this particular paper on bottom-up processes during single word reading, on which there is already a lot to say.

To clarify our focus on feed-forward process, we have modified the title of the paper to be:

“Convolutional networks can model the functional modulation of the MEG responses associated with feed-forward processes during visual word recognition” furthermore, we have revised the Introduction to highlight this choice, noting:

“Another limitation is that these models have primarily focused on feed-back lexicosemantic effects while oversimplifying the initial feed-forward processing of the visual input.

[…]

For this study, we chose to focus on modeling the early feed-forward processing occurring during visual word recognition, as the experimental setup in Vartiainen et al. (2011) was designed to demonstrate.

[…]

By doing so, we restrict ourselves to an investigation of how well the three evoked components can be explained by a feed-forward CNN in an experimental setting designed to demonstrate feed-forward effects. As such, the goal is not to present a complete model of all aspects of reading, which should include feed-back effects, but rather to demonstrate the effectiveness of using a model that has a realistic form of input when the aim is to align the model with the evoked responses observed during visual word recognition.”

And in the Discussion section:

“In this paper we have restricted our simulations to feed-forward processes. Now, the way is open to incorporate convolution-and-pooling principles in models of reading that simulate feed-back processes as well, which should allow the model to capture more nuance in the Type-II and N400m components, as well as extend the simulation to encompass a realistic semantic representation.”

(2) There is no clear relationship between the layers that signal needs to traverse in the model and the relative duration of the three components in the brain.

While some models offer a tentative mapping between layers and locations in the brain, none of the models we are aware of actually simulate time accurately and our model is no exception.

While we provide some evidence that the three MEG components are best modeled with different types of layers, and the type-I becomes somewhere before type-II and N400m is last in our model, the lack of timing information is a weakness of our model we have not been able to address. In our previous version, this already was the main topic of our “Limitations of the model” section, but since this weakness was pointed out by all reviewers, we have decided to widen our discussion of it:

“One important limitation of the current model is the lack of an explicit mapping from the units inside its layers to specific locations in the brain at specific times. The temporal ordering of the components is simulated correctly, with the response profile matching that of the type-I occurring the layers before those matching the type-II, followed by the N400m. Furthermore, every component is best modeled by a different type of layer, with the type-I best described by convolution-and-pooling, the type-II by fully-connected linear layers and the N400m by a one-hot encoded layer. However, there is no clear relationship between the number of layers the signal needs to traverse in the model to the processing time in the brain. Even if one considers that the operations performed by the initial two convolution layers happen in the retina rather than the brain, the signal needs to propagate through three more convolution layers to reach the point where it matches the type-II component at 140-200 ms, but only through one more additional layer to reach the point where it starts to match the N400m component at 300-500 ms. Still, cutting down on the number of times convolution is performed in the model seems to make it unable to achieve the desired suppression of noise (Figure 5). It also raises the question what the brain is doing during the time between the type-II and N400m component that seems to take so long. It is possible that the timings of the MEG components are not indicative solely of when the feed-forward signal first reaches a certain location, but are rather dictated by the resolution of feed-forward and feedback signals (Nour Eddine et al., 2024).”

See also our response to the next comment of the Reviewer, in which we dive more into the effect of the number of layers, which could be seen as a manipulation of time.

(3) I am impressed by the CNN that authors modified to match the human brain pattern for the visual word recognition process, by the increase and decrease of the number of layers. The result of this part was a little different from the author’s expectation; however, the author didn’t explain or address this issue.

We are glad to hear that the reviewer found these results interesting. Accordingly, we now discuss these results more thoroughly in the main text.

We have moved the figure from the supplementary information to the main text (Figure 5 in the revised manuscript). And describe the results in the Results section:

“Various variations in model architecture and training procedure were evaluated. We found that the number of layers had a large impact on the response patterns produced by the model (Figure 5). The original VGG-11 architecture defines 5 convolution layers and 3 fully connected layers (including the output layer). Removing a convolution layer (Figure 5, top row), or removing one of the fully connected layers (Figure 5, second row), resulted in a model that did exhibit an enlarged response to noisy stimuli in the early layers that mimics the Type-I response. However, such models failed to show a sufficiently diminished response to noisy stimuli in the later layers, hence failing to produce responses that mimic the Type-II or N400m, a failure which also showed as low correlation scores.

Adding an additional convolution layer (Figure 5, third row) resulted in a model where none of the layer response profiles mimics that of the Type-II response. The Type-II response is characterized by a reduced response to both noise and symbols, but an equally large response to consonant strings, real and pseudo words. However, in the model with an additional convolution layer, the consonant strings evoked a reduced response already in the first fully connected layer, which is a feature of the N400m rather than the Type-II. These kind of subtleties in the response pattern, which are important for the qualitative analysis, generally did not show quantitatively in the correlation scores, as the fully connected layers in this model correlate as well with the Type-II response as models that did show a response pattern that mimics the Type-II.

Adding an additional fully connected layer (Figure 5, fourth row) resulted in a model with similar response profiles and correlation with the MEG components as the original VGG-11 architecture (Figure 5, bottom row) The N400m-like response profile is now observed in the third fully connected layer rather than the output layer. However, the decrease in response to consonant strings versus real and pseudo words, which is typical of the N400m, is less distinct than in the original VGG-11 architecture.”

We also incorporated these results in the Discussion:

“However, the ability of pooling units to stratify such differences depends on the size of their receptive field, which is determined by the number of convolution-andpooling layers. This might also explain why, in later layers, we observed a decreased response to stimuli where text was rendered with a font size exceeding the receptive field of the pooling units (Figure 8). Hence, the response profiles of the subsequent fully connected layers was very sensitive to the number of convolution-and-pooling layers. This number is probably dependent on the input size and pooling strategy. Given the VGG11 design of doubling the receptive field after each layer, combined with an input size of 225x225 pixels, the optimal number of convolution-and-pooling layers for our model was five, or the model would struggle to produce response profiles mimicking those of the type-II component in the subsequent fully connected layers (Figure 5).

[…]

A minimum of two fully connected layers was needed to achieve this in our case, and adding more fully connected layers would make them behave more like the component (Figure 5).”

(4) Can the author explain why the number of layers in the final model is optimal by benchmarking the brain hierarchy?

We have incorporated the figure describing the correlation between each model and the MEG components (previously Figure 5) with the figures describing the response profiles (Figures 4 and 5 in the revised manuscript and Supplementary Figures 2-6). This way, we (and the reader) can now benchmark every model qualitatively and quantitatively.

As we stated in our response to the previous comment, we have added a more thorough discussion on the number of layers, which includes the justification for our choice for the final model. The benchmark we used was primarily whether the model shows the same response patterns as the Type I, Type II and N400 responses, which disqualifies all models with fewer than 5 convolution and 3 fully connected layers. Models with more layers also show the proper response patterns, however we see that there is actually very little difference in the correlation scores between different models. Hence, our justification for sticking with the original VGG11 architecture is that it produces the qualitative best response profiles, while having roughly the same (decently high) correlation with the MEG components. Furthermore, by sticking to the standard architecture, we make it slightly easier to replicate our results as one can use readily available pre-trained ImageNet weights.

As well as always discussing the correlation scores in tandem with the qualitative analysis, we have added the following statement to the Results:

“Based on our qualitative and quantitative analysis, the model variant that performed best overall was the model that had the original VGG11 architecture and was preinitialized from earlier training on ImageNet, as depicted in the bottom rows of Figure 4 and Figure 5.”

**Reviewer #2 (Public Review):**
As has been shown over many decades, many potential computational algorithms, with varied model architectures, can perform the task of text recognition from an image. However, there is no evidence presented here that this particular algorithm has comparable performance to human behavior (i.e. similar accuracy with a comparable pattern of mistakes). This is a fundamental prerequisite before attempting to meaningfully correlate these layer activations to human neural activations. Therefore, it is unlikely that correlating these derived layer weights to neural activity provides meaningful novel insights into neural computation beyond what is seen using traditional experimental methods.

We very much agree with the reviewer that a qualitative analysis of whether the model can explain experimental effects needs to happen before a quantitative analysis, such as evaluating model-brain correlation scores. In fact, this is one of the intended key points we wished to make.

As we discuss at length in the Introduction, “traditional” models of reading (those that do not rely on deep learning) are not able to recognize a word regardless of exact letter shape, size, and (up to a point) rotation. In this study, our focus is on these low-level visual tasks rather than high-level tasks concerning semantics. As the Reviewer correctly states, there are many potential computational algorithms able to perform these visual task at a human level and so we need to evaluate the model not only on its ability to mimic human accuracy but also on generating a comparable pattern of mistakes. In our case, we need a pattern of behavior that is indicative of the visual processes at the beginning of the reading pipeline. Hence, rather than relying on behavioral responses that are produced at the very end, we chose the evaluate the model based on three MEG components that provide “snapshots” of the reading process at various stages. These components are known to manifest a distinct pattern of “behavior” in the way they respond to different experimental conditions (Figure 2), akin to what to Reviewer refers to as a “pattern of mistakes”. The model was first evaluated on its ability to replicate the behavior of the MEG components in a qualitative manner (Figure 4). Only then do we move on to a quantitative correlation analysis. In this manner, we feel we are in agreement with the approach advocated by the Reviewer.

In the Introduction, we now clarify:

“Another limitation is that these models have primarily focused on feed-back lexicosemantic effects while oversimplifying the initial feed-forward processing of the visual input.

[…]

We sought to construct a model that is able to recognize words regardless of length, size, typeface and rotation, as well as humans can, so essentially perfectly, whilst producing activity that mimics the type-I, type-II, and N400m components which serve as snapshots of this process unfolding in the brain.

[…]

These variations were first evaluated on their ability to replicate the experimental effects in that study, namely that the type-I response is larger for noise embedded words than all other stimuli, the type-II response is larger for all letter strings than symbols, and that the N400m is larger for real and pseudowords than consonant strings. Once a variation was found that could reproduce these effects satisfactorily, it was further evaluated based on the correlation between the amount of activation of the units in the model and MEG response amplitude.”

To make this prerequisite more clear, we have removed what was previously Figure 5, which showed the correlation between the various models the MEG components out of the context of their response patterns. Instead, these correlation values are now always presented next to the response patterns (Figures 4 and 5, and Supplementary Figures 2-6 in the revised manuscript). This invites the reader to always consider these metrics in relation to one another.

One example of a substantial discrepancy between this model and neural activations is that, while incorporating frequency weighting into the training data is shown to slightly increase neural correlation with the model, Figure 7 shows that no layer of the model appears directly sensitive to word frequency. This is in stark contrast to the strong neural sensitivity to word frequency seen in EEG (e.g. Dambacher et al 2006 Brain Research), fMRI (e.g. Kronbichler et al 2004 NeuroImage), MEG (e.g. Huizeling et al 2021 Neurobio. Lang.), and intracranial (e.g. Woolnough et al 2022 J. Neurosci.) recordings. Figure 7 also demonstrates that the late stages of the model show a strong negative correlation with font size, whereas later stages of neural visual word processing are typically insensitive to differences in visual features, instead showing sensitivity to lexical factors.

We are glad the reviewer brought up the topic of frequency balancing, as it is a good example of the importance of the qualitative analysis. Frequency balancing during training only had a moderate impact on correlation scores and from that point of view does not seem impactful. However, when we look at the qualitative evaluation, we see that with a large vocabulary, a model without frequency balancing fails to properly distinguish between consonant strings and (pseudo)words (Figure 4, 5th row). Hence, from the point of view of being able to reproduce experimental effects, frequency balancing had a large impact. We now discuss this more explicitly in the revised Discussion section:

“Overall, we found that a qualitative evaluation of the response profiles was more helpful than correlation scores. Often, a deficit in the response profile of a layer that would cause a decrease in correlation on one condition would be masked by an increased correlation in another condition. A notable example is the necessity for frequency-balancing the training data when building models with a vocabulary of 10 000. Going by correlation score alone, there does not seem to be much difference between the model trained with and without frequency balancing (Figure 4A, fifth row versus bottom row). However, without frequency balancing, we found that the model did not show a response profile where consonant strings were distinguished from words and pseudowords (Figure 4A, fifth row), which is an important behavioral trait that sets the N400m component apart from the Type-II component (Figure 2D). This underlines the importance of the qualitative evaluation in this study, which was only possible because of a straightforward link between the activity simulated within a model to measurements obtained from the brain, combined with the presence of clear experimental conditions.”

It is true that the model, even with frequency balancing, only captures letter- and bigramfrequency effects and not the word-frequency effects that we know the N400m is sensitive to. Since our model is restricted to feed-forward processes, this finding adds to the evidence that frequency-modulated effects are driven by feed-back effects as modeled by Nour Eddine et al. (2024, doi:10.1016/j.cognition.2024.105755). See also our response to the next comment by the Reviewer where we discuss feed-back connections. We have added the following to the section about model limitations in the revised Discussion:

“The fact that the model failed to simulate the effects of word-frequency on the N400m (Figure 8), even after frequency-balancing of the training data, is additional evidence that this effect may be driven by feed-back activity, as for example modeled by Nour Eddine et al. (2024).”

Like the Reviewer, we initially thought that later stages of neural visual word processing would be insensitive to differences in font size. When diving into the literature to find support for this claim, we found only a few works directly studying the effect of font size on evoked responses, but, surprisingly, what we did find seemed to align with our model. We have added the following to our revised Discussion:

“The fully connected linear layers in the model show a negative correlation with font size. While the N400 has been shown to be unaffected by font size during repetition priming (Chauncey et al., 2008), it has been shown that in the absence of priming, larger font sizes decrease the evoked activity in the 300–500 ms window (Bayer et al., 2012; Schindler et al., 2018). Those studies refer to the activity within this time window, which seems to encompass the N400, as early posterior negativity (EPN). What possibly happens in the model is that an increase in font size causes an initial stronger activation in the first layers, due to more convolution units receiving input. This leads to a better signal-to-noise ratio (SNR) later on, as the noise added to the activation of the units remains constant whilst the amplitude of the input signal increases. A better SNR translates ultimately in less co-activation of units corresponding to orthographic neighbours in the final layers, hence to a decrease in overall layer activity.”

Another example of the mismatch between this model and the visual cortex is the lack of feedback connections in the model. Within the visual cortex, there are extensive feedback connections, with later processing stages providing recursive feedback to earlier stages. This is especially evident in reading, where feedback from lexical-level processes feeds back to letter-level processes (e.g. Heilbron et al 2020 Nature Comms.). This feedback is especially relevant for the reading of words in noisy conditions, as tested in the current manuscript, as lexical knowledge enhances letter representation in the visual cortex (the word superiority effect). This results in neural activity in multiple cortical areas varying over time, changing selectivity within a region at different measured time points (e.g. Woolnough et al 2021 Nature Human Behav.), which in the current study is simplified down to three discrete time windows, each attributed to different spatial locations.

We agree with the Reviewer that a full model of reading in the brain must include feed-back connections and share their sentiment that these feed-back processes play an important role and are a fascinating topic to study. The intent for the model presented in our study is very much to be a stepping stone towards extending the capabilities of models that do include such connections.

However, there is a problem of scale that cannot be ignored.

Current models of reading that do include feedback connections fall into the category we refer to in the paper as “traditional models” and all only a few layers deep and operate on very simplified inputs, such as pre-defined line segments, a few pixels, or even a list of prerecognized letters. The Heilbron et al. 2020 study that the Reviewer refers to is a good example of such a model. (This excellent and relevant work was somehow overlooked in our literature discussion in the Introduction. We thank the Reviewer for pointing it out to us.) Models incorporating realistic feed-back activity need these simplifications, because they have a tendency to no longer converge when there are too many layers and units. However, in order for models of reading to be able to simulate cognitive behavior such as resolving variations in font size or typeface, or distinguish text from non-text, they need to operate on something close to the pixel-level data, which means they need many layers and units.

Hence, as a stepping stone, it is reasonable to evaluate a model that has the necessary scale, but lacks the feed-back connections that would be problematic at this scale, to see what it can and cannot do in terms of explaining experimental effects in neuroimaging studies. This was the intended scope of our study. For the revision, we have attempted to make this more clear.

We have changed the title to be:

“Convolutional networks can model the functional modulation of the MEG responses associated with feed-forward processes during visual word recognition” and added the following to the Introduction:

“The simulated environments in these models are extremely simplified, partly due to computational limitations and partly due to the complex interaction of feed-forward and feed-back connectivity that causes problems with convergence when the model grows too large. Consequently, these models have primarily focused on feed-back lexico-semantic effects while oversimplifying the initial feed-forward processing of the visual input.

[…]

This rather high level of visual representation sidesteps having to deal with issues such as visual noise, letters with different scales, rotations and fonts, segmentation of the individual letters, and so on. More importantly, it makes it impossible to create the visual noise and symbol string conditions used in the MEG study to modulate the type-I and type-II components. In order to model the process of visual word recognition to the extent where one may reproduce neuroimaging studies such as Vartiainen et al. (2011), we need to start with a model of vision that is able to directly operate on the pixels of a stimulus. We sought to construct a model that is able to recognize words regardless of length, size, typeface and rotation with very high accuracy, whilst producing activity that mimics the type-I, type-II, and N400m components which serve as snapshots of this process unfolding in the brain. For this model, we chose to focus on the early feed-forward processing occurring during visual word recognition, as the experimental setup in the MEG study was designed to demonstrate, rather than feed-back effects

[…]

By doing so, we restrict ourselves to an investigation of how well the three evoked components can be explained by a feed-forward CNN in an experimental setting designed to demonstrate feed-forward effects. > As such, the goal is not to present a complete model of all aspects of reading, which should include feed-back effects, but rather to demonstrate the effectiveness of using a model that has a realistic form of input when the aim is to align the model with the evoked responses observed during visual word recognition.”

And we have added the following to the Discussion section:

“In this paper we have restricted our simulations to feed-forward processes. Now, the way is open to incorporate convolution-and-pooling principles in models of reading that simulate feed-back processes as well, which should allow the model to capture more nuance in the Type-II and N400m components, as well as extend the simulation to encompass a realistic semantic representation. A promising way forward may be to use a network architecture like CORNet (Kubilius et al., 2019), that performs convolution multiple times in a recurrent fashion, yet simultaneously propagates activity forward after each pass. The introduction of recursion into the model will furthermore align it better with traditional-style models, since it can cause a model to exhibit attractor behavior (McLeod et al., 2000), which will be especially important when extending the model into the semantic domain.

Furthermore, convolution-and-pooling has recently been explored in the domain of predictive coding models (Ororbia & Mali, 2023), a type of model that seems particularly well suited to model feed-back processes during reading (Gagl et al., 2020; Heilbron et al., 2020; Nour Eddine et al., 2024).”

We also would like to point out to the Reviewer that we did in fact perform a correlation between the model and the MNE-dSPM source estimate of all cortical locations and timepoints (Figure 7B). Such a brain-wide correlation map confirms that the three dipole groups are excellent summaries of when and where interesting effects occur within this dataset.

The presented model needs substantial further development to be able to replicate, both behaviorally and neurally, many of the well-characterized phenomena seen in human behavior and neural recordings that are fundamental hallmarks of human visual word processing. Until that point, it is unclear what novel contributions can be gleaned from correlating low-dimensional model weights from these computational models with human neural data.

We hope that our revisions have clarified the goals and scope of this study. The CNN model we present in this study is a small but, we feel, essential piece in a bigger effort to employ deep learning techniques to further enhance already existing models of reading. In our revision, we have extended our discussion where to go from here and outline our vision on how these techniques could help us better model the phenomena the reviewer speaks of. We agree with the reviewer that there is a long way to go, and we are excited to be a part of it.

In addition to the changes described above, we now end the Discussion section as follows:

“Despite its limitations, our model is an important milestone for computational models of reading that leverages deep learning techniques to encompass the entire computational process starting from raw pixels values to representations of wordforms in the mental lexicon. The overall goal is to work towards models that can reproduce the dynamics observed in brain activity observed during the large number of neuroimaging experiments performed with human volunteers that have been performed over the last few decades. To achieve this, models need to be able to operate on more realistic inputs than a collection of predefined lines or letter banks (for example: Coltheart et al., 2001; Heilbron et al., 2020; Laszlo & Armstrong, 2014; McClelland & Rumelhart, 1981; Nour Eddine et al., 2024). We have shown that even without feed-back connections, a CNN can simulate the behavior of three important MEG evoked components across a range of experimental conditions, but only if unit activations are noisy and the frequency of occurrence of words in the training dataset mimics their frequency of use in actual language.”

**Reviewer #3 (Public Review):**
The paper is rather qualitative in nature. In particular, the authors show that some resemblance exists between the behavior of some layers and some parts of the brain, but it is hard to quantitively understand how strong the resemblances are in each layer, and the exact impact of experimental settings such as the frequency balancing (which seems to only have a very moderate effect according to Figure 5).

The large focus on a qualitative evaluation of the model is intentional. The ability of the model to reproduce experimental effects (Figure 4) is a pre-requisite for any subsequent quantitative metrics (such as correlation) to be valid. The introduction of frequency balancing is a good example of this. As the reviewer points out, frequency balancing during training has only a moderate impact on correlation scores and from that point of view does not seem impactful. However, when we look at the qualitative evaluation, we see that with a large vocabulary, a model without frequency balancing fails to properly distinguish between consonant strings and (pseudo)words (Figure 4, 5th row). Hence, from the point of view of being able to reproduce experimental effects, frequency balancing has a large impact.

That said, the reviewer is right to highlight the value of quantitative analysis. An important limitation of the “traditional” models of reading that do not employ deep learning is that they operate in unrealistically simplified environments (e.g. input as predefined line segments, words of a fixed length), which makes a quantitative comparison with brain data problematic. The main benefit that deep learning brings may very well be the increase in scale that makes more direct comparisons with brain data possible. In our revision we attempt to capitalize on this benefit more. The reviewer has provided some helpful suggestions for doing so in their recommendations, which we discuss in detail below.

We have added the following discussion on the topic of qualitative versus quantitative analysis to the Introduction:

“We sought to construct a model that is able to recognize words regardless of length, size, typeface and rotation, as well as humans can, so essentially perfectly, whilst producing activity that mimics the type-I, type-II, and N400m components which serve as snapshots of this process unfolding in the brain.

[…]

These variations were first evaluated on their ability to replicate the experimental effects in that study, namely that the type-I response is larger for noise embedded words than all other stimuli, the type-II response is larger for all letter strings than symbols, and that the N400m is larger for real and pseudowords than consonant strings. Once a variation was found that could reproduce these effects satisfactorily, it was further evaluated based on the correlation between the amount of activation of the units in the model and MEG response amplitude.”

And follow this up in the Discussion with a new sub-section entitled “On the importance of experimental contrasts and qualitative analysis of the model”

The experiments only consider a rather outdated vision model (VGG).

VGG was designed to use a minimal number of operations (convolution-and-pooling, fullyconnected linear steps, ReLU activations, and batch normalization) and rely mostly on scale to solve the classification task. This makes VGG a good place to start our explorations and see how far a basic CNN can take us in terms of explaining experimental MEG effects in visual word recognition. However, we agree with the reviewer that it is easy to envision more advanced models that could potentially explain more. In our revision, we expand on the question of where to go from here and outline our vision on what types of models would be worth investigating and how one may go about doing that in a way that provides insights beyond higher correlation values.

We have included the following in our Discussion sub-sections on “Limitations of the current model and the path forward”:

“The VGG-11 architecture was originally designed to achieve high image classification accuracy on the ImageNet challenge (Simonyan & Zisserman, 2015). Although we have introduced some modifications that make the model more biologically plausible, the final model is still incomplete in many ways as a complete model of brain function during reading.

[…]

In this paper we have restricted our simulations to feed-forward processes. Now, the way is open to incorporate convolution-and-pooling principles in models of reading that simulate feed-back processes as well, which should allow the model to capture more nuance in the Type-II and N400m components, as well as extend the simulation to encompass a realistic semantic representation. A promising way forward may be to use a network architecture like CORNet (Kubilius et al., 2019), that performs convolution multiple times in a recurrent fashion, yet simultaneously propagates activity forward after each pass. The introduction of recursion into the model will furthermore align it better with traditional-style models, since it can cause a model to exhibit attractor behavior (McLeod et al., 2000), which will be especially important when extending the model into the semantic domain. Furthermore, convolution-and-pooling has recently been explored in the domain of predictive coding models (Ororbia & Mali, 2023), a type of model that seems particularly well suited to model feed-back processes during reading (Gagl et al., 2020; Heilbron et al., 2020; Nour Eddine et al., 2024).”

**Reviewer #3 (Recommendations For The Authors):**
(1) The method used to select the experimental conditions under which the behavior of the CNN is the most brain-like is rather qualitative (Figure 4). It would have been nice to have a plot where the noisyness of the activations, the vocab size and the amount of frequency balancing are varied continuously, and show how these three parameters impact the correlation of the model layers with the MEG responses.

We now include this analysis (Figure 6 in the revised manuscript, Supplementary Figures 47) and discuss these factors in the revised Results section:

“Various other aspects of the model architecture were evaluated which ultimately did not lead to any improvements of the model. The response profiles can be found in the supplementary information (Supplementary Figures 4–7) and the correlations between the models and the MEG components are presented in Figure 6. The vocabulary of the final model (10 000) exceeds the number of units in its fullyconnected layers, which means that a bottleneck is created in which a sub-lexical representation is formed. The number of units in the fully-connected layers, i.e. the width of the bottleneck, has some effect on the correlation between model and brain (Figure 6A), and the amount of noise added to the unit activations less so (Figure 6B). We already saw that the size of the vocabulary, i.e. the number of wordforms in the training data and number of units in the output layer of the model, had a large effect on the response profiles (Figure 4). Having a large vocabulary is of course desirable from a functional point of view, but also modestly improves correlation between model and brain (Figure 6C). For large vocabularies, we found it beneficial to apply frequency-balancing of the training data, meaning that the number of times a word-form appears in the training data is scaled according to its frequency in a large text corpus. However, this cannot be a one-to-one scaling, since the most frequent words occur so much more often than other words that the training data would consist of mostly the top-ten most common words, with less common words only occurring once or not at all. Therefore, we decided to scale not by the frequency 𝑓 directly, but by 𝑓𝑠, where 0 < 𝑠 < 1, opting for 𝑠 = 0.2 for the final model (Figure 6D).”

(2) It is not clear which layers exactly correspond to which of the three response components. For this to be clearer, it would have been nice to have a plot with all the layers of VGG on the x-axis and three curves corresponding to the correlation of each layer with each of the three response components.

This is a great suggestion that we were happy to incorporate in the revised version of the manuscript. Every figure comparing the response patterns of the model and brain now includes a panel depicting the correlation between each layer of the model and each of the three MEG components (Figures 4 & 5, Supplementary Figures 2-5). This has given us (and now also the reader) the ability to better benchmark the different models quantitatively, adding to our discussion on qualitative to quantitative analysis.

(3) It is not clear to me why the authors report the correlation of all layers with the MEG responses in Figure 5: why not only report the correlation of the final layers for N400, and that of the first layers for type-I?

We agree with the reviewer that it would have been better to compare the correlation scores for those layers which response profile matches the MEG component. While the old Figure 5 has been merged with Figure 4, and now provides the correlations between all the layers and all MEG components, we have taken the Reviewer’s advice and marked the layers which qualitatively best correspond to each MEG component, so the reader can take that into account when interpreting the correlation scores.

(4) The authors mention that the reason that they did not reproduce the protocol with more advanced vision models is that they needed the minimal setup capable of yielding the desired experiment effect. I am not fully convinced by this and think the paper could be significantly strengthened by reporting results for a vision transformer, in particular to study the role of attention layers which are expected to play an important role in processing higher-level features.

We appreciate and share the Reviewer’s enthusiasm in seeing how other model architectures would fare when it comes to modeling MEG components. However, we regard modifying the core model architecture (i.e., a series of convolution-and-pooling followed by fully-connected layers) to be out of scope for the current paper.

One of the key points of our study is to create a model that reproduces the experimental effects of an existing MEG study, which necessitates modeling the initial feed-forward processing from pixel to word-form. For this purpose, a convolution-and-pooling model was the obvious choice, because these operations play a big role in cognitive models of vision in general. In order to properly capture all experimental contrasts in the MEG study, many variations of the CNN were trained and evaluated. This iterative design process concluded when all experimental contrasts could be faithfully reproduced.

If we were to explore different model architectures, such as a transformer architecture, reproducing the experimental contrasts of the MEG study would no longer be the end goal, and it would be unclear what the end goal should be. Maximizing correlation scores has no end, and there are a nearly endless number of model architectures one could try. We could bring in a second MEG study with experimental contrasts that the CNN cannot explain and a transformer architecture potentially could and set the end goal to explain all experimental effects in both MEG studies. But even if we had access to such a dataset, this would almost double the length of the paper, which is already too long.